# PRIORITIZED SOFT Q-DECOMPOSITION FOR LEXICOGRAPHIC REINFORCEMENT LEARNING

**Finn Rietz**[*]
Örebro University
Sweden

**Erik Schaffernicht**
Örebro University
Sweden

**Stefan Heinrich**
IT University of Copenhagen
Denmark

**Johannes A. Stork**
Örebro University
Sweden

## ABSTRACT

Reinforcement learning (RL) for complex tasks remains a challenge, primarily due to the difficulties of engineering scalar reward functions and the inherent inefficiency of training models from scratch. Instead, it would be better to specify complex tasks in terms of elementary subtasks and to reuse subtask solutions whenever possible. In this work, we address continuous space lexicographic multi-objective RL problems, consisting of prioritized subtasks, which are notoriously difficult to solve. We show that these can be scalarized with a subtask transformation and then solved incrementally using value decomposition. Exploiting this insight, we propose *prioritized soft Q-decomposition* (PSQD), a novel algorithm for learning and adapting subtask solutions under lexicographic priorities in continuous state-action spaces. PSQD offers the ability to reuse previously learned subtask solutions in a zero-shot composition, followed by an adaptation step. Its ability to use retained subtask training data for offline learning eliminates the need for new environment interaction during adaptation. We demonstrate the efficacy of our approach by presenting successful learning, reuse, and adaptation results for both low- and high-dimensional simulated robot control tasks, as well as offline learning results. In contrast to baseline approaches, PSQD does not trade off between conflicting subtasks or priority constraints and satisfies subtask priorities during learning. PSQD provides an intuitive framework for tackling complex RL problems, offering insights into the inner workings of the subtask composition.

## 1    INTRODUCTION

Reinforcement learning (RL) for complex tasks is challenging because as practitioners we must design a scalar-valued reward function that induces the desired behavior (Ha et al., 2021; Tessler et al., 2019; Irpan, 2018) and we must expensively learn each task from scratch. Instead, it might be more suitable to model complex tasks as several simpler subtasks (Gábor et al., 1998; Roijers et al., 2013; Nguyen et al., 2020; Hayes et al., 2022), and more efficient to reuse or adapt subtask solutions that were learned previously (Arnekvist et al., 2019; Hausman et al., 2018; Finn et al., 2017). To prevent potentially conflicting subtasks from deteriorating the overall solution, we can impose lexicographic priorities on the subtasks (Skalse et al., 2022; Gábor et al., 1998; Zhang et al., 2022). This means ordering all subtasks by priority and forbidding lower-priority subtasks from worsening the performance of higher-priority subtasks, which is similar to null-space projection approaches for complex multi-task control problems (Dietrich et al., 2015). In RL, this setting is referred to as a *lexicographic* multi-objective reinforcement learning (MORL) problem.

However, state-of-the-art RL algorithms for continuous state and action spaces like Soft Q-Learning (SQL) (Haarnoja et al., 2017), Soft Actor-Critic (SAC) (Haarnoja et al., 2018b), or Proximal Policy Optimization (PPO) (Schulman et al., 2017) do not support lexicographic MORL problems out of the box and obvious extensions still trade off conflicting subtasks, leading to suboptimal solutions, as we demonstrate in our experiments. Current lexicographic MORL algorithms like (Li & Czarnecki, 2019; Zhang et al., 2022) enumerate actions and are therefore limited to discrete problems or rely on constrained optimization methods that trade-off between optimizing subtask and priority objectives (Skalse et al., 2022). All these lexicographic algorithms learn a monolithic policy,

---

[*]Corresponding author: `finn.rietz@oru.se`

precluding any interpretation or reuse of previously learned subtask solutions. Furthermore, it is unknown what overall reward function these approaches actually optimize, which hampers principled understanding of lexicographic MORL algorithms.

In this paper, we are the first to show that such complex, lexicographic MORL problems can be scalarized (Roijers et al., 2013) and we propose *prioritized soft Q-decomposition* (PSQD), a learning algorithm that solves lexicographic MORL tasks in a decomposed fashion (Russell & Zimdars, 2003). For this, we proposed a subtask transformation that recovers the Q-function of the lexicographic MORL problem as the sum of Q-functions of the transformed subtasks and so provides the overall scalar reward function. PSQD computes this sum incrementally by learning and transforming subtask Q-functions, beginning at the highest-priority subtask and step-by-step learning and adding Q-functions for lower-priority subtasks. In every step, an arbiter ensures that the lower-priority subtasks do not worsen already solved higher-priority subtasks and restricts the lower-priority subtask to solutions within the so-called *action indifference space* of the higher-priority subtasks. PSQD provides a safe learning and exploration framework, which is crucial for many real-world scenarios, and allows reuse and adaptation of separately learned subtask solutions to more complex, lexicographic MORL task, which saves data and computation costs. The decomposable Q-function benefits interpretability of the learned behavior by allowing analysis and inspection of subtask solutions and their action indifference spaces.

The contributions of this paper are: (1) We show that lexicographic MORL problems can be scalarized using our proposed subtask transformation. (2) We show how to reuse and adapt separately learned subtask solutions to more complex, lexicographic MORL problems, starting from a priority-respecting zero-shot composition solution. (3) We provide PSQD, an algorithm for lexicographic MORL problems with continuous state-action spaces, that reuses both subtask solutions and retained offline data.

## 2 PROBLEM DEFINITION AND PRELIMINARIES

We formalize the complex lexicographic MORL problem (Skalse et al., 2022; Gábor et al., 1998) as a Markov decision process (MDP) $\mathcal{M} \equiv (\mathcal{S}, \mathcal{A}, \mathbf{r}, p_s, \gamma, \succ)$ consisting of state space $\mathcal{S}$, action space $\mathcal{A}$, reward function $\mathbf{r} \colon \mathcal{S} \times \mathcal{A} \to \mathbb{R}^n$, and discount factor $\gamma \in [0, 1]$. The subtasks are numbered $i = 1, 2, \ldots n$ and ranked with the transitive priority relation $1 \succ 2 \succ \ldots n$ with task 1 having the highest and task $n$ having the lowest priority. Each dimension of the vector-valued reward corresponds to one scalar subtask reward function $[\mathbf{r}(\mathbf{s}, \mathbf{a})]_i = r_i(\mathbf{s}, \mathbf{a})$. The discrete-time dynamics are given by a conditional distribution $p_s(\mathbf{s}_{t+1} \mid \mathbf{s}_t, \mathbf{a}_t)$ with $\mathbf{s}_t, \mathbf{s}_{t+1} \in \mathcal{S}$ and $\mathbf{a}_t \in \mathcal{A}$. We use $(\mathbf{s}_t, \mathbf{a}_t) \sim \rho_\pi$ to denote the state-action marginal induced by a policy $\pi(\mathbf{a}_t \mid \mathbf{s}_t)$.

Lexicographic subtask priorities can be defined by constraining the policy in each subtask $i$ to stay close to optimal (i.e., $\varepsilon$-optimal) for the next higher-priority task $i - 1$ (Skalse et al., 2022; Zhang et al., 2022). Formally, such an $\varepsilon$-lexicographic MORL problem implies a set of allowed policies $\Pi_i$ for each subtask $i$,

$$\Pi_i = \{\pi \in \Pi_{i-1} \mid \max_{\pi' \in \Pi_{i-1}} J_{i-1}(\pi') - J_{i-1}(\pi) \le \varepsilon_{i-1}\}, \tag{1}$$

where $J_i$ are performance criteria like value-functions, and $0 \le \varepsilon_i \in \mathbb{R}$ are thresholds for each subtask. However, it is not practical to compute the sets $\Pi_i$ explicitly when solving $\varepsilon$-lexicographic MORL problems.

Current approaches solve $\varepsilon$-lexicographic MORL problems with constrained RL or incremental action selection in discrete domains (Skalse et al., 2022; Zhang et al., 2022) and it is unclear what equivalent scalar reward function these methods actually optimize (Skalse et al., 2022; Gábor et al., 1998; Vamplew et al., 2022). In contrast, we transform the individual subtasks and their Q-functions to implement the priority constraints in equation 1. This leads to an equivalent scalar reward function for the complex $\varepsilon$-lexicographic MORL problem. Our definition of the performance criteria $J_i$ in Sec. 3.1 is key to our incremental and decomposed learning approach with this reward function.

### 2.1 Q-DECOMPOSITION

Our goal is to reuse and adapt subtask solutions that were learned previously to new and more complex $\varepsilon$-lexicographic MORL problems. For this, we require a decomposed learning algorithm

that models individual subtask solutions and combines them to the solution of the complex $\varepsilon$-lexicographic MORL problem.

*Q-decomposition* (Russell & Zimdars, 2003) is such an algorithm for MORL problems where the complex task's reward function is the sum of subtask rewards, $r_\Sigma(\mathbf{s}_t, \mathbf{a}_t) = \sum_{i=1}^n [\mathbf{r}(\mathbf{s}_t, \mathbf{a}_t)]_i$. Instead of learning the sum Q-function, $Q_\Sigma$, the algorithm learns one Q-function $Q_i$ for each subtask and submits them to a central *arbiter*. The arbiter combines the Q-functions, $Q_\Sigma(\mathbf{s}, \mathbf{a}) = \sum_{i=1}^n Q_i(\mathbf{s}, \mathbf{a})$, and selects the maximizing action for its policy $\pi_\Sigma$. Q-decomposition uses on-policy SARSA-style (Sutton & Barto, 2018; Rummery & Niranjan, 1994) updates with the arbiter policy $\pi_\Sigma$ to avoid *illusion of control* (Russell & Zimdars, 2003) or the *attractor phenomenon* Laroche et al. (2017); van Seijen et al. (2017) in subtask learning.

Summing and scaling subtask rewards alone does not result in lexicographic priorities because low-priority subtasks with sufficiently large reward scales can still dictate over high-priority subtasks. Therefore, we cannot directly use the Q-decomposition algorithm for lexicographic MORL. Our algorithm builds on Q-decomposition but we transform subtasks to ensure that reward- and Q-functions can be summed up *whilst* implementing lexicographic priorities. To emphasize this difference, we denote transformed subtask Q-functions by $Q_{\succ i}$, which estimate the long-term value of transformed subtask rewards $r_{\succ i}$, such that the Q-function of the lexicographic MORL problem corresponds to $Q_{1 \succ 2 \succ \dots n}(\mathbf{s}, \mathbf{a}) = \sum_{i=1}^n Q_{\succ i}(\mathbf{s}, \mathbf{a})$, or $Q_\succ$ for short. $Q_i$ and $r_i$ refer to the untransformed subtask Q-functions and rewards in the lexicographic MORL problem.

## 2.2 MAXIMUM ENTROPY REINFORCEMENT LEARNING

We aim to solve $\varepsilon$-lexicographic MORL problems with continuous states and actions, but instead of explicitly computing equation 1, we focus on identifying as many *near optimal actions* as possible, such that lower-priority subtasks have more alternative actions to choose from. For this reason, we use maximum entropy reinforcement learning (MaxEnt RL) (Ziebart et al., 2008; Haarnoja et al., 2018b; 2017) for learning subtask policies $\pi_i$ and subtask Q-functions $Q_i$. MaxEnt RL maximizes the reward signal augmented by the Shannon entropy of the policy's action distribution, $\mathcal{H}(\mathbf{a}_t \mid \mathbf{s}_t) = \mathbb{E}_{\mathbf{a}_t \sim \pi(\mathbf{a}_t \mid \mathbf{s}_t)}[-\log \pi(\mathbf{a}_t \mid \mathbf{s}_t)]$, i.e.,

$$\pi_{\text{MaxEnt}}^* = \arg\max_\pi \sum_{t=1}^\infty \mathbb{E}_{(\mathbf{s}_t, \mathbf{a}_t) \sim \rho_\pi}\big[\gamma^{t-1}\big(r(\mathbf{s}_t, \mathbf{a}_t) + \alpha \mathcal{H}(\mathbf{a}_t \mid \mathbf{s}_t)\big)\big], \tag{2}$$

where the scaler $\alpha$ can be used to determine the relative importance of the entropy term. This improves exploration in multimodal problems but also prevents overly deterministic policies, which is key for identifying many near-optimal actions. Following Haarnoja et al. (2017), we drop $\alpha$ because it can be replaced with reward scaling. Importantly, the entropy term in the objective leads to the following energy-based Boltzmann distribution as the optimal MaxEnt policy,

$$\pi_{\text{MaxEnt}}^*(\mathbf{a}_t \mid \mathbf{s}_t) = \exp\big(Q_{\text{soft}}^*(\mathbf{s}_t, \mathbf{a}_t) - V_{\text{soft}}^*(\mathbf{s}_t)\big), \tag{3}$$

where $Q_{\text{soft}}^*(\mathbf{s}_t, \mathbf{a}_t)$ acts as the negative energy and $V_{\text{soft}}^*(\mathbf{s}_t)$ is the log-partition function. The *soft* Q-function $Q_{\text{soft}}^*$ and *soft* value function $V_{\text{soft}}^*$ in equation 3 are given by

$$Q_{\text{soft}}^*(\mathbf{s}_t, \mathbf{a}_t) = r(\mathbf{s}_t, \mathbf{a}_t) + \mathbb{E}_{(\mathbf{s}_{t+1}, \dots) \sim \rho_\pi}\bigg[\sum_{l=1}^\infty \gamma^l \big(r(\mathbf{s}_{t+l}, \mathbf{a}_{t+l}) + \mathcal{H}(\mathbf{a}_{t+l} \mid \mathbf{s}_{t+l})\big)\bigg], \tag{4}$$

$$V_{\text{soft}}^*(\mathbf{s}_t) = \log \int_{\mathcal{A}} \exp(Q_{\text{soft}}^*(\mathbf{s}_t, \mathbf{a}')) \, d\mathbf{a}', \tag{5}$$

where the entropy in equation 4 is based on the optimal MaxEnt policy $\pi_{\text{MaxEnt}}^*$. For sampling actions, we can disregard the log-partition function in equation 3 and directly exploit the proportionality relationship,

$$\pi_{\text{MaxEnt}}^*(\mathbf{a}_t \mid \mathbf{s}_t) \propto \exp\big(Q_{\text{soft}}^*(\mathbf{s}_t, \mathbf{a}_t)\big), \tag{6}$$

between the policy and the soft Q-function, e.g., with Monte Carlo methods (Metropolis et al., 1953; Hastings, 1970; Duane et al., 1987). We drop the $_{\text{MaxEnt}}$ and $_{\text{soft}}$ subscripts for the remainder of the paper to avoid visual clutter. The *soft Q-learning* (Haarnoja et al., 2017) algorithm implements a *soft* Bellman contraction operator and in practice learns a model for sampling the complex Boltzmann distribution in equation 6. Our algorithm, PSQD, builds on soft Q-learning and combines it with $\varepsilon$-lexicographic priorities as well as Q-decomposition from above.

## 3 PRIORITIZED SOFT Q-DECOMPOSITION

In this section, we explain our approach for solving $\varepsilon$-lexicographic MORL problems. In Sec. 3.1, we explain our subtask transformation for modeling $\varepsilon$-lexicographic priorities and show how it scalarizes $\varepsilon$-lexicographic MORL problems. In Sec. 3.2, we derive our decomposed learning algorithm, PSQD, for $\varepsilon$-lexicographic MORL problems and describe a practical version of this algorithm for continuous spaces in Sec. 3.3.

### 3.1 SUBTASK TRANSFORMATION FOR LEXICOGRAPHIC PRIORITIES IN MAXENT RL

Our goal is to learn a MaxEnt (arbiter) policy $\pi_\succ(\mathbf{a} \mid \mathbf{s}) \propto \exp(Q_\succ(\mathbf{s}, \mathbf{a}))$ for the $\varepsilon$-lexicographic MORL problem that fulfills the constraint in equation 1. However, instead of explicitly representing the sets $\Pi_i$, we focus on the action space and define a state-based version of the $\varepsilon$-optimality constraint on the policy $\pi_\succ$ using the on-arbiter subtask Q-functions $Q_i$

$$\max_{\mathbf{a}' \in \mathcal{A}} Q_i(\mathbf{s}, \mathbf{a}') - Q_i(\mathbf{s}, \mathbf{a}) \le \varepsilon_i, \forall \mathbf{a} \sim \pi_\succ, \forall \mathbf{s} \in \mathcal{S}, \forall i \in \{1, \dots, n-1\}. \tag{7}$$

This restricts $\pi_\succ$ in each state to actions that are close to the optimal for each subtask $i < n$. As discussed in Sec. 2.1, summing up subtask Q-functions and using the proportionality relationship in equation 6 to directly define an arbiter policy $\pi_\Sigma$ will in general *not* satisfy the $\varepsilon$-lexicographic priority constraints in equation 7. Therefore, we propose a transformation of the subtasks and their Q-functions which provides that the constraints are satisfied when the *transformed* subtask Q-functions are summed up to $Q_\succ$. Our transformation relies on a constraint indicator function $c_i \colon \mathcal{S} \times \mathcal{A} \to \{0, 1\}$ that shows whether an action $\mathbf{a}$ in state $\mathbf{s}$ is permitted by the priority constraint for subtasks $i$,

$$c_i(\mathbf{s}, \mathbf{a}) = \begin{cases} 1, & \max_{\mathbf{a}' \in \mathcal{A}} Q_i(\mathbf{s}, \mathbf{a}') - Q_i(\mathbf{s}, \mathbf{a}) \le \varepsilon_i \\ 0, & \text{otherwise} \end{cases}. \tag{8}$$

We refer to the set of permitted actions by subtask $i$ in state $\mathbf{s}$, $\mathcal{A}_{\succ i}(\mathbf{s}) = \{\mathbf{a} \in \mathcal{A} \mid c_i(\mathbf{s}, \mathbf{a}) = 1\}$, as the *action indifference space* of subtask $i$. The intuition is that subtask $i$ is indifferent as to which $\varepsilon$-optimal action in $\mathcal{A}_{\succ i}(\mathbf{s})$ is selected because they are all close enough to optimal. Analogously, the intersection $\mathcal{A}_\succ(\mathbf{s}) = \cap_{i=1}^{n-1} \mathcal{A}_{\succ i}(\mathbf{s})$ is the global action indifference space and $\bar{\mathcal{A}}_\succ(\mathbf{s}) = \mathcal{A} \setminus \mathcal{A}_\succ(\mathbf{s})$ is the set of forbidden actions. As mentioned above, keeping $\mathcal{A}_{\succ i}(\mathbf{s})$ large is crucial because it provides more actions for optimizing the arbiter policy.

Using $Q_i$ and $c_i$ as defined above, we can now write down the Q-function of the $\varepsilon$-lexicographic MORL problem, which shows that constraint-violating actions are forbidden by their Q-values,

$$Q_\succ(\mathbf{s}, \mathbf{a}) = \sum_{i=1}^{n-1} \ln(c_i(\mathbf{s}, \mathbf{a})) + Q_n(\mathbf{s}, \mathbf{a}), \tag{9}$$

where we use $\ln(1) = 0$ and $\lim_{x \to 0^+} \ln(x) = -\infty$. Here, the $\ln(c_i)$ terms are the transformed Q-functions for subtasks $i < n$ in the view of Q-decomposition. Moreover, $Q_\succ$ is equal to $Q_n$ for all $\mathbf{a} \in \mathcal{A}_\succ(\mathbf{s})$ and $Q_\succ(\mathbf{s}, \mathbf{a}) = -\infty$ otherwise. Our MaxEnt arbiter policy $\pi_\succ(\mathbf{a} \mid \mathbf{s}) \propto \exp(Q_\succ(\mathbf{s}, \mathbf{a}))$ has a product structure,

$$\pi_\succ(\mathbf{a} \mid \mathbf{s}) \propto \exp\left(\sum_{i=1}^{n-1} \ln(c_i(\mathbf{s}, \mathbf{a})) + Q_n(\mathbf{s}, \mathbf{a})\right) = \left(\prod_{i=1}^{n-1} c_i(\mathbf{s}, \mathbf{a})\right) \exp Q_n(\mathbf{s}, \mathbf{a}), \tag{10}$$

confirming that $\pi_\succ$ has likelihood 0 for actions outside the global indifference space whilst selecting actions inside the global indifference spaces proportional to $\exp Q_n$.

A close inspection of equation 9 shows that the corresponding transformed subtask reward functions are $r_{\succ i}(\mathbf{s}, \mathbf{a}) = \ln(c_i(\mathbf{s}, \mathbf{a}))$ and $r_{\succ n} = r_n$, further revealing $r_\succ = \sum_{i=1}^{n} r_{\succ i}$ as the scalar reward function of the $\varepsilon$-lexicographic MORL problem. For a detailed derivation, see supplementary material in Sec. A. The Q-decomposition view suggests a decomposed learning algorithm with the transformed rewards $r_{\succ i}$. However, because the transformed rewards $r_{\succ i}$ are defined as the transformed Q-functions $Q_{\succ i}$, our decomposed algorithm learns $Q_i$ and obtains $Q_{\succ i}$ by first computing $c_i$ for equation 9. Furthermore, $Q_i$ are learned on-policy for the arbiter, to avoid illusion of control in the subtasks (Russell & Zimdars, 2003; Laroche et al., 2017; van Seijen et al., 2017).

### 3.2 Incremental and decomposed learning with lexicographic priorities

We want to exploit the structure of the Q-function $Q_\succ$ and policy $\pi_\succ$ of the $\varepsilon$-lexicographic MORL problem as described in Sec. 3.1 to define an incremental and decomposed learning approach that can reuse and adapt subtask solutions. Our algorithm incrementally includes more subtasks ordered by their priority: In the first step, we learn subtask 1 (with highest priority) without any restrictions. In the following steps, we learn subtask $1 < i \leq n$ restricted to the action indifference spaces of subtasks with higher priority than $i$ and reuse the results of higher-priority subtasks, i.e. $Q_1$ to $Q_{i-1}$ for the components of $Q_\succ$. Subtasks with priority lower than $i$ do not affect the learning of subtask $i$. In this and the following sections, we describe the process generically for the last subtask $i = n$ because it aligns with the notation in equation 9.

In the Q-decomposition view, we are optimizing the arbiter policy $\pi_\succ$ with Q-function $Q_\succ$ for the MORL problem with (transformed) rewards $r_{\succ 1}, \ldots, r_{\succ n-1}$ and we are given on-arbiter-policy estimates $Q_i^{\pi_\succ}$ for subtasks $1 \leq i < n$. As seen in equation 9, our means for improving the arbiter policy $\pi_\succ$ is maximizing the term $Q_n$ for subtask $n$. For this, we perform a *policy evaluation step* under the arbiter policy to get a new estimate of the subtask Q-function $Q_n^{\pi_\succ}$, which also changes our estimate of the global arbiter Q-function $Q_\succ$—we call this *arbiter policy evaluation*. Next, we perform a *policy improvement step* by softly maximizing $Q_n^{\pi_\succ}$, which again also changes $Q_\succ$—we call this *arbiter policy improvement*. This algorithm closely resembles the Q-decomposition algorithm in Sec. 2.1 because $Q_n$ is updated on-policy with the global arbiter policy $\pi_\succ$. We theoretically analyze a *subtask* view of this algorithm below and provide an additional *arbiter* view analysis and interpretation of this algorithm in supplementary material Sec. B.

**Subtask view theory.**    In the subtask view, the algorithm optimizes subtask $n$ with soft Q-learning in an MDP $\mathcal{M}_{\succ n}$ that is like $\mathcal{M}$ in Sec. 2 but uses scalar reward $r_n$ and has the action space restricted to the global action indifference space, $\mathcal{A}_\succ(\mathbf{s})$, in every state. Therefore, the action space of $\mathcal{M}_{\succ n}$ satisfies the lexicographic priority constraints by construction. This allows us to perform off-policy soft Q-learning in $\mathcal{M}_{\succ n}$ to directly obtain $Q_{\succ n}^*$, while still respecting all priority constraints. We do this with an *off-policy, soft Bellman backup operator* $\mathcal{T}$

$$\mathcal{T}Q(\mathbf{s}, \mathbf{a}) \triangleq r(\mathbf{s}, \mathbf{a}) + \gamma \mathbb{E}_{\mathbf{s}' \sim p} \left[ \underbrace{\log \int_{\mathcal{A}_\succ(\mathbf{s}')} \exp\left(Q(\mathbf{s}', \mathbf{a}')\right) d\mathbf{a}'}_{V(\mathbf{s}')} \right], \tag{11}$$

which is the update step from soft Q-learning (Haarnoja et al., 2017) and where $\mathbf{a} \in \mathcal{A}_\succ(\mathbf{s})$.

**Theorem 3.1** (Prioritized Soft Q-learning). *Consider the soft Bellman backup operator $\mathcal{T}$, and an initial mapping $Q^0 : \mathcal{S} \times \mathcal{A}_\succ \to \mathbb{R}$ with $|\mathcal{A}_\succ| < \infty$ and define $Q^{l+1} = \mathcal{T}Q^l$, then the sequence of $Q^l$ converges to $Q_\succ^*$, the soft Q-value of the optimal arbiter policy $\pi_\succ^*$, as $l \to \infty$.*

*Proof.* Proof and detailed derivation in supplementary material Sec. F.4. □

While the described backup operator $\mathcal{T}$ yields the optimal global solution in tabular settings, it is intractable in large and continuous state and action spaces. In the next section, we convert $\mathcal{T}$ and Theorem 3.1 into a stochastic optimization problem that can be solved approximately via parametric function approximation.

### 3.3 Practical learning and adaptation algorithm for continuous spaces

For PSQD, we implement the subtask view described in Sec. 3.2, building on the soft Q-learning (SQL) algorithm (Haarnoja et al., 2017). Learning subtask $n$ in the MDP $\mathcal{M}_{\succ n}$ requires an efficient way for sampling in the global action indifference space $\mathcal{A}_\succ(\mathbf{s})$, which becomes increasingly difficult in higher dimensions due to the curse of dimensionality. However, we can exploit the incrementally learned subtask policies $\pi_i(\mathbf{a} \mid \mathbf{s}) \propto \exp Q_i(\mathbf{a}, \mathbf{s})$ to efficiently sample in subtask action indifference spaces because the high-probability regions of subtask policies *are* the subtask indifference spaces. For rejecting action samples that are outside of the global action indifference space, $\mathbf{a} \notin \mathcal{A}_\succ(\mathbf{s})$, we evaluate the constraint indicator functions $c_i$ in equation 8. The remaining action samples are used in importance sampling proportionally to $\exp Q_n(\mathbf{a}, \mathbf{s})$.

Technically, we learn $\pi_n$ and $Q_n$ for the subtask $n$ by integrating the sampling process from above in SQL, which corresponds to performing soft Q-learning in $\mathcal{M}_{\succ n}$. We model the subtask Q-function by a DNN $Q_n$ with parameters $\theta$ and minimizes the temporal-difference loss

$$J_Q(\theta) = \mathbb{E}_{\mathbf{s}_t, \mathbf{a}_t \sim \mathcal{D}} \left[ \frac{1}{2} \Big( Q_n^\theta(\mathbf{s}_t, \mathbf{a}_t) - r_n(\mathbf{s}_t, \mathbf{a}_t) + \gamma \mathbb{E}_{\mathbf{s}_{t+1} \sim p} \big[ V_n^{\bar{\theta}}(\mathbf{s}_{t+1}) \big] \Big)^2 \right], \qquad (12)$$

where $\bar{\theta}$ is a target parameter obtained as exponentially moving average of $\theta$ and $V_n^{\bar{\theta}}$ is the empirical approximation of equation 5. The subtask policy is represented by a state-conditioned, stochastic DNN $f_n$ with parameters $\phi$, such that $\mathbf{a}_t = f_n^\phi(\zeta_t, \mathbf{s}_t)$, where $\zeta_t$ represents noise from an arbitrary distribution. The parameter $\phi$ is updated by minimizing the (empirical approximation of the) Kullback-Leibler divergence between $f_n^\phi$ and the optimal MaxEnt arbiter policy in equation 3:

$$J_\pi(\phi) = \mathrm{D}_{\mathrm{KL}} \Big( f_n^\phi(\cdot | \mathbf{s}_t, \zeta_t) \,\Big\|\, \exp \big( Q_\succ^\theta(\mathbf{s}_t, \cdot) - V_\succ^\theta(\mathbf{s}_t) \big) \Big). \qquad (13)$$

The objectives $J_\pi(\phi)$ and $J_Q(\theta)$ can be computed offline, given a replay buffer of saved transitions. Thus, PSQD can optimize the $n$-th subtask offline, for example reusing the transitions collected during independent and isolated training for subtask $n$ in the subsequent adaptation step for subtask $n$ as part of the $\varepsilon$-lexicographic MORL problem, as detailed in supplementary material C. A pictographic overview of our method as well as pseudocode can be found in supplementary material D.

## 4 EXPERIMENTS

We evaluate PSQD in an instructive 2D navigation environment and a high-dimensional control problem (details in Sec. G in the supplementary material). In Sec. 4.1 and Sec. 4.2, we qualitatively analyze zero-shot and offline adaptation results, respectively. In Sec. 4.3, we empirically compare our method with a number of baseline methods. Finally, in Sec. 4.4, we demonstrate PSQD's efficacy in a high-dimensional, continuous control environment. For all experiments, we first learn the (untransformed) subtasks individually such that we have access to DNN approximations of $Q_i^*$ and $\pi_i^*$ for reuse and adaptation. We use the hat symbol, e.g., $\hat{\pi}$, to denote DNN approximations.

### 4.1 ZERO-SHOT REUSE OF PREVIOUSLY LEARNED SUBTASKS SOLUTIONS

In this experiment, we demonstrate that our approach provides solutions that satisfy lexicographic priorities even in the zero-shot setting without interacting with the new environment. We design a 2D navigation environment (see Fig. 1a), where the agent is initialized at random positions and has to reach the goal area at the top while avoiding a $\cap$-shaped obstacle. For the zero-shot solution, we simply apply our transformation (Sec. 3.1) to the already learned subtask Q-functions $\hat{Q}_1^*, \hat{Q}_2^*$ to obtain the Q-function $Q_{1 \succ 2} = \ln(c_1) + \hat{Q}_2^*$. The modular structure of $Q_{1 \succ 2}$ allows us to inspect the components of the zero-shot solution: Fig. 1b reveals that $\hat{Q}_1^*$ learns the obstacle avoidance subtask with large negative values inside the obstacle and inspecting the $\ln(c_1)$ component shows our transformation and how it implements the priority constraint. In Fig. 1c, we plot $\ln(c_1((\mathbf{s}, \mathbf{a})), \forall \mathbf{a} \in \mathcal{A}$, with the agent placed at the red dot in Fig. 1a to show the action indifference space at that state. Our transformation excludes actions that would lead to collisions, allowing us to visually infer the agent's behavior. This is in stark contrast to prior zero-shot methods, e.g. (Haarnoja et al., 2018a), where the behavior of the zero-shot solution is unpredictable and potentially arbitrary sub-optimal, as noted by Hunt et al. (2019). In Fig. 1d, we visualize the policy $\pi_{1 \succ 2} \propto Q_{1 \succ 2}$ with sample trajectories and project $\ln(c_1)$ onto the image to illustrate the global action indifference space. Our zero-shot solution never collides with the obstacle due to the priority constraint without any training on the lexicographic MORL problem. However, our zero-shot is suboptimal for the lexicographic MORL problem, because it sometimes gets trapped and fails to reach the goal for starting positions in the middle. In Sec. G (supplementary material), we show more detailed zero-shot results.

### 4.2 OFFLINE ADAPTATION AND DATA REUSE

In this experiment, we demonstrate that PSQD can adapt previously learned subtask solutions to the lexicographic MORL problem using only the training data from the previous, isolated learning process without environment interaction. After adapting $\hat{Q}_2^*$ with the offline data, $\hat{Q}_2^{\pi_\succ}$ no longer

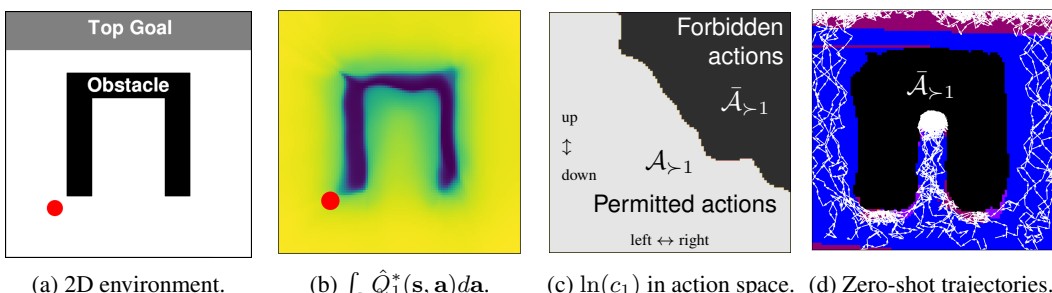

(a) 2D environment.     (b) $\int_{\mathbf{a}} \hat{Q}_1^*(\mathbf{s}, \mathbf{a})d\mathbf{a}$.     (c) $\ln(c_1)$ in action space.    (d) Zero-shot trajectories.

Figure 1: **Zero-shot experiment** in the 2D navigation environment. $\hat{Q}_1^*$ in 1b (brighter hues indicate higher value) and its transformed version in 1c (evaluated at red dot) forbid actions that lead to obstacle collisions. Sample traces in 1d (larger version in Fig. 6) show navigation towards the goal at the top, sometimes getting stuck but without colliding with the obstacle. The background in 1d is colored according to discretized angles of the policy ↖[-1, 1] ↑[0, 1] ↗[1, 1] ←[-1, 0] →[1, 0] ↙[-1, -1] ↓[0, -1] ↘[1, -1].

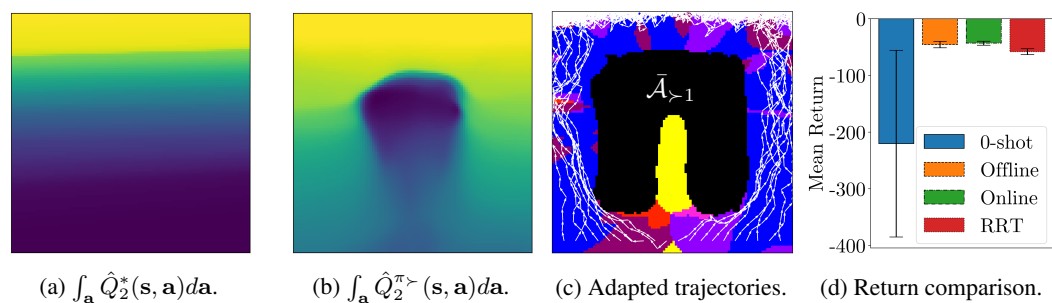

(a) $\int_{\mathbf{a}} \hat{Q}_2^*(\mathbf{s}, \mathbf{a})d\mathbf{a}$.     (b) $\int_{\mathbf{a}} \hat{Q}_2^{\pi_\succ}(\mathbf{s}, \mathbf{a})d\mathbf{a}$.     (c) Adapted trajectories.    (d) Return comparison.

Figure 2: **Offline adaptation experiment** in 2D navigation environment. Our learning algorithm adapts the pre-trained $\hat{Q}_2^*$ in 2a to $\hat{Q}_2^{\pi_\succ}$ in 2b (brighter hues indicate higher value), reflecting the long-term value of $r_2$ under the arbiter policy. The adapted agent has learned to drive out of and around the obstacle, as shown in 2c (larger version in Fig. 6). The background in 2c is colored in the same way as 1d. Both online and offline adaptation improve upon the zero-shot agent considerably, as shown in 2d.

reflects the greedy policy $\pi_2^*$ that gets the zero-shot solution stuck, but instead reflects the arbiter agent $\pi_\succ^*$ and accounts for the long-term outcome. Comparing $\hat{Q}_2^*$ in Fig. 2a with $\hat{Q}_2^{\pi_\succ}$ in Fig. 2b shows that the adaptation process leads to lower values inside the ∩-shape such that our solution escapes and avoids that region. This optimal behavior is illustrated with sample traces in Fig. 2c. In Fig. 2d we compare the mean return of the zero-shot, offline, and online adapted agents and an RRT (LaValle & Kuffner Jr, 2001; Karaman & Frazzoli, 2010) oracle. This shows that the zero-shot agent, which gets stuck, performs considerably worse than the adapted agents, which instead are on par with the RRT oracle. We continue this analysis for the more complex tasks $r_{1\succ2\succ3}$ and $r_{1\succ3\succ2}$ in supplementary material Sec. G.

### 4.3 EMPIRICAL COMPARISON AGAINST BASELINES

In this experiment, we compare PSQD to a set of five baseline methods which all try to implement lexicographic priorities in MORL problems. As, to the best of our knowledge, PSQD is the first method for solving lexicographic MORL problems with continuous action spaces, we compare against obvious extensions of established RL algorithms and ablations of PSQD: Naive subtask-priority implementation for SAC (Haarnoja et al., 2018b) and PPO (Schulman et al., 2017), a modified version of Q-Decomposition (Russell & Zimdars, 2003) for continuous action spaces that we call *soft Q-decomposition* (SQD), as well as the SQL composition method proposed by Haarnoja et al. (2018a). For the ablation of PSQD, instead of transforming subtasks, we introduce additional negative reward for priority constraint violations based on the constraint indicator functions $c_i$. For SAC and PPO we naively implement subtask-priorities by augmenting the learning objective to additionally include the KL-divergence between the current and the previously learned higher-priority policy. SQD concurrently learns the on-arbiter-policy Q-functions for all subtasks. The SQL compo-

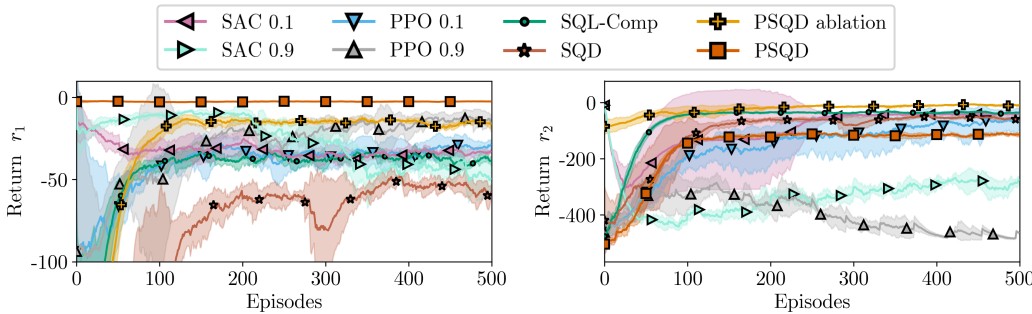

Figure 3: **Baseline comparison** in the 2D navigation environment. **Left**: Cost of the high-priority obstacle avoidance subtask during learning of the lower priority task. **Right**: Lower-priority navigation cost. For SAC and PPO, the scalar in the legend refers to the weight $\beta$ of the convex policy objective, where $\beta = 1$ places all weight on the KL term.

sition (Haarnoja et al., 2018a) also learns solutions for all subtasks concurrently, however off-policy and thereby with illusion of control. See Sec. G.3 for more details on the baseline methods.

For SAC, PPO, and PSQD (incl. ablation), we first train all algorithms on the obstacle avoidance subtask 1, where all methods converge to an approximately optimal solution. Subsequently, we reuse the obtained $Q_1$ and train each method on subtask 2, recording the returns for both subtasks separately in Fig. 3. PSQD is the only method that maintains zero return for subtask 1 (i.e. zero collisions) while learning for subtask 2, outperforming all other baselines. As seen in the right panel of Fig. 3, PSQD obtains less reward for subtask 2 than other baselines. However, this is because the priority constraint prevents PSQD from navigating through the obstacle, while some baselines accept obstacle collisions to reach the goal quicker. In contrast to PSQD, none of the baseline methods converge to a solution that satisfies the lexicographic priorities and maintains the optimal performance in the top-priority collision avoidance subtask that they initially achieved.

### 4.4 HIGH-DIMENSIONAL CONTROL

In this experiment, we demonstrate that PSQD scales well to high-dimensional action spaces while maintaining subtask priorities, unlike traditionally composed multi-objective agents. We simulated an Franka Emika Panda joint-control task with a 9-dimensional action space (Fig 4, left), where the high-priority subtasks wants to avoid the red cube while the low-priority subtask want to reach the green sphere with the end-effector. We start with the zero-shot Q-function $Q_{1 \succ 2} = \ln(c_1) + \hat{Q}_2^*$ and sample from $\pi_\succ$ as described in Sec. 3.3. We subsequently adapt the zero-shot solution to obtain $Q_{1 \succ 2}^* = \ln(c_1) + \hat{Q}_2^{\pi_\succ}$. We also include the SQL-Comp. method (Haarnoja et al., 2018a) as a baseline, by composing $Q_{1+2} = \hat{Q}_1^* + \hat{Q}_2^*$. Representative trajectories from our adapted solution and the baseline are shown in Fig. 4 (left). Starting from $t = 0$ in the leftmost panel, our adapted agent moves around the obstacle. In contrast, the baseline violates the priority and moves through the forbidden part of the workspace. More quantitatively, we show mean returns for the different agents in Fig. 4 (right). Our solution avoids the volume even in the zero-shot setting and is improved considerably through the adaptation process, while the baseline induces high costs in $r_1$ because it moves through the forbidden part of the workspace. We note that Haarnoja et al. (2018a) do not claim that their method solves lexicographic MORL problems, we use it to illustrate that priorities are generally difficult to express in traditional multi-objective tasks.

## 5 RELATED WORK

**Task priorities for continuous action-spaces.** To the best of our knowledge, PSQD is the first method to solve general lexicographic MORL tasks with continuous action-spaces. The following prior works consider task priorities in RL: Yang et al. (2021) combine null-space control and RL by constraining policy search to the null-space of higher priority constraints. However, Yang et al. (2021)'s method is a special case of PSQD since it requires access to the task-space Jacobian, while PSQD exploits given Q-functions. Skalse et al. (2022); Zhang et al. (2022); Li & Czarnecki (2019) also model lexicographic task priorities, however only in discrete settings and without revealing the reward function of the lexicographic task. PSQD generalizes these methods to continuous action-

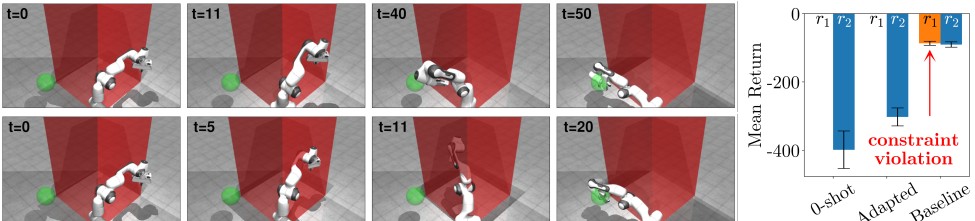

Figure 4: **High-dimensional joint control** using a simulated Franka Emika Panda robot. **Top left**: Our prioritized agent respects constraints even in the zero-shot setting and is improved by our learning algorithm. **Bottom left**: A multi-objective baseline Haarnoja et al. (2018a) does not respect lexicographic priorities and generates high costs in $r_1$ (collision) in favor of fewer costs in $r_2$ (trajectory length), moving through the forbidden part of the workspace.

spaces and maximizes a known reward function. Furthermore, none of these works emphasize composition and reuse of subtask solutions, which is central to our method.

**Composition for knowledge transfer.** Composition for transfer between tasks in the MaxEnt framework has initially been demonstrated by Haarnoja et al. (2018a), however, only in a zero-shot setting without providing a learning algorithm to improve the composed agent and without priorities. Hunt et al. (2019) build upon Haarnoja et al. (2018a) and propose a learning algorithm that learns the divergence between the subtask Q-functions and obtains the optimal solution to convex task compositions. Thus, similar to our method, these works facilitate knowledge transfer between tasks via composition, however, they offer no way for modeling strict task priorities.

**Task priorities for safety.** Task priorities can be used to implement safety. Prior works mostly implement (safety-) constrained RL via constrained MDPs (CMDPs) (Altman, 1999), where policy search is constrained by an *unordered* set of cost functions. CMDPs can be solved by optimizing Lagrangian objectives and dual gradient descent techniques Boyd et al. (2004). This way, (Ha et al., 2021; Tessler et al., 2019; Chow et al., 2017) focus on CMDPs with one handcrafted cost function, while (Achiam et al., 2017; Liang et al., 2018) provide methods for solving CMPDs with multiple constraints. While these methods yield constraint-satisfying agents, unlike our method, they do not facilitate reuse and adaptation via composition, since they optimize monolithic policies from scratch. An exception w.r.t reuse is *safety critic* (Srinivasan et al., 2020), which pre-trains a transferable estimator of expected future safety incidents. However, safety-critic can only incorporate one such critic and thus only transfer knowledge from one pre-training task, while our method can exploit any number of pre-trained Q-functions.

## 6 LIMITATIONS

Our method depends on the manual selection of $\varepsilon_1, \ldots, \varepsilon_{n-1}$ thresholds. These scalars are on the scale of subtask Q-functions $Q_i$, which might be hard to estimate when function approximators are employed and depend on domain-specific reward functions, user preferences, and task semantics. There are, however, a number of informed ways for finding adequate $\varepsilon_i$ scalars. In practice, one can analyze the Q-function in key-states and select $\varepsilon_i$ such that undesired actions are excluded. This works even when the action space is of high dimensionality, since approximate Q-function mean, min, max, and percentiles can be computed via sampling.

## 7 CONCLUSION

The main contribution of this paper is a principled framework for solving lexicographic MORL problems with continuous action spaces in a decomposed fashion. In this framework, we can zero-shot interpretable agents that respect task priority constraints in low- and high-dimensional settings. Our learning algorithm, PSQD, facilitates reuse of subtask solutions by adapting them to solve the lexicographic MORL problem optimally.

ACKNOWLEDGMENTS

This work was partially supported by the Wallenberg AI, Autonomous Systems and Software Program (WASP) funded by the Knut and Alice Wallenberg Foundation.

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

SUPPLEMENTARY MATERIAL

# A    LEXICOGRAPHIC MULTI-OBJECTIVE TASKS

Here, we describe in detail how we model lexicographic task-priority constraints as scalarizable multi-objective RL tasks. To recap, we have multiple reward functions $r_1, \ldots, r_n$, and we want to find a global arbiter policy $\pi_\succ$ that maximizes $r_n$, while behaving almost optimal w.r.t. all higher priority tasks $r_1, \ldots, r_{n-1}$. This is expressed with the constraint

$$\max_{\mathbf{a}' \in \mathcal{A}} Q_i(\mathbf{s}, \mathbf{a}') - Q_i(\mathbf{s}, \mathbf{a}) \leq \varepsilon_i, \forall \mathbf{a} \sim \pi, \forall \mathbf{s} \in \mathcal{S}, \forall i \in \{1, \ldots, n-1\} \tag{14}$$

on the performance measure $J(\pi)$, which in this case is the Q-value. We note again the indicator functions

$$c_i(\mathbf{s}, \mathbf{a}) = \begin{cases} 1, & \max_{\mathbf{a}' \in \mathcal{A}} Q_i(\mathbf{s}, \mathbf{a}') - Q_i(\mathbf{s}, \mathbf{a}) \leq \varepsilon_i \\ 0, & \text{otherwise} \end{cases}, \tag{15}$$

and that $\prod_{i=1}^{n-1} c_i(\mathbf{s}, \mathbf{a}) = 1$ when an action $\mathbf{a}$ is allowed in state $\mathbf{s}$ according to each of the higher priority constraints (and thus in the global action indifference space $\mathcal{A}_\succ(\mathbf{s})$). The key idea now is to transform the higher-priority Q-functions $Q_i$ into new reward functions $r_{\succ i}$ by applying the natural logarithm to the constraint indicator functions, such that $r_{\succ i}(\mathbf{s}, \mathbf{a}) = \ln(c_i(\mathbf{s}, \mathbf{a}))$. Based on these transformed Q-functions and new rewards, we now define a multi-objective, vector-valued reward function that reflects strict task priorities

$$\mathbf{r}_\succ(\mathbf{s}, \mathbf{a}) = \begin{bmatrix} r_{\succ 1}(\mathbf{s}, \mathbf{a}) \\ r_{\succ 2}(\mathbf{s}, \mathbf{a}) \\ \vdots \\ r_{\succ n-1}(\mathbf{s}, \mathbf{a}) \\ r_n(\mathbf{s}, \mathbf{a}) \end{bmatrix} = \begin{bmatrix} \ln(c_1(\mathbf{s}, \mathbf{a})) \\ \ln(c_2(\mathbf{s}, \mathbf{a})) \\ \vdots \\ \ln(c_{n-1}(\mathbf{s}, \mathbf{a})) \\ r_n(\mathbf{s}, \mathbf{a}) \end{bmatrix}. \tag{16}$$

We define the global reward function that the arbiter maximizes as sum of transformed rewards

$$r_\succ(\mathbf{s}, \mathbf{a}) = \sum_{i=1}^{n} [\mathbf{r}(\mathbf{s}, \mathbf{a})]_i = \underbrace{\sum_{i=1}^{n-1} \ln c_i(\mathbf{s}, \mathbf{a})}_{\text{constraint indication}} + r_n(\mathbf{s}, \mathbf{a}), \tag{17}$$

such that we are in the Q-decomposition setting Russell & Zimdars (2003). This allows us to decompose the learning problem into $n$ subtask Q-functions $Q_1, \ldots, Q_n$ and one global policy $\pi_\succ$. The global Q-function $Q_\succ$ for the prioritized task $r_\succ$ corresponds to the sum of subtask Q-functions, $Q_\succ = \sum_{i=1}^{n-1} Q_{\succ i} + Q_n$, where the first $n-1$ higher-priority Q-functions are for the transformed rewards, while the $n$-th Q-function is for the ordinary reward function $r_n$. Because of MaxEnt RL the global arbiter policy $\pi_\succ$ satisfies the proportional relationship $\pi_\succ(\mathbf{a}|\mathbf{s}) \propto \exp(Q_\succ(\mathbf{s}, \mathbf{a}))$, as described in Sec. 2.2.

The global reward function in equation 17 allows us to infer certain properties of $Q_\succ$ and $\pi_\succ$. Firstly, by noting that $\ln(1) = 0$ we see that the constraint indication summation in equation 17 is zero when action $\mathbf{a}$ is allowed according to all higher priority tasks and in the global action indifference space $\mathcal{A}_\succ(\mathbf{s})$. This means that $r_\succ(\mathbf{s}, \mathbf{a}) = r_n(\mathbf{s}, \mathbf{a}), \forall \mathbf{s} \in \mathcal{S}, \forall \mathbf{a} \in \mathcal{A}_\succ(\mathbf{s})$, i.e. maximizing the global task corresponds to maximizing the lowest priority tasks in the global indifference space, which is a property we exploit extensively.

Next, by defining $\lim_{x \to 0^+} \ln(x) = -\infty$, we see that $r_\succ(\mathbf{s}, \mathbf{a}) = -\infty$ when any number of higher priority constraints are violated by $\mathbf{a}$ in $\mathbf{s}$. Furthermore, by noting that $-\infty \pm \mathbb{R} = -\infty$ and $-\infty \cdot \mathbb{R}_{\geq 0} = -\infty$, we see that $Q_\succ(\mathbf{s}, \mathbf{a}) = -\infty, \forall \mathbf{s} \in \mathcal{S}, \forall \mathbf{a} \in \bar{\mathcal{A}}_\succ(\mathbf{s})$, i.e. the value of any constraint-violating action is $-\infty$ under the global task $r_\succ$. Importantly, due to the proportional relationship $\pi_\succ(\mathbf{a}|\mathbf{s}) \propto \exp(Q_\succ(\mathbf{s}, \mathbf{a}))$, it follows that such constraint-violating actions have zero probability under our arbiter policies. This also means that the expected future constraint-violation value of arbiter policies is zero, since the arbiter policy can not select constraint-violating actions in the future. With this in mind, for the global Q-function of arbiter policies, we see that

$$
\begin{aligned}
Q_{\succ}(\mathbf{s}, \mathbf{a}) &= \sum_{i=1}^{n} Q_{\succ i}(\mathbf{s}, \mathbf{a}) \\
&= \sum_{i=1}^{n-1} Q_{\succ i}(\mathbf{s}, \mathbf{a}) + Q_n(\mathbf{s}, \mathbf{a}) \\
&= \sum_{i=1}^{n-1} r_{\succ i}(\mathbf{s}, \mathbf{a}) + r_n(\mathbf{s}, \mathbf{a}) + \gamma \mathbb{E}_{\mathbf{s}_{t+1} \sim p, \mathbf{a}_{t+1} \sim \rho_{\pi_{\succ}}} \big[ \overbrace{Q_{\succ i}(\mathbf{s}_{t+1}, \mathbf{a}_{t+1})}^{0 \text{ due to } \pi_{\succ}} + Q_n(\mathbf{s}_{t+1}, \mathbf{a}_{t+1}) \big] \\
&= \sum_{i=1}^{n-1} r_{\succ i}(\mathbf{s}, \mathbf{a}) + r_n(\mathbf{s}, \mathbf{a}) + \gamma \mathbb{E}_{\mathbf{s}_{t+1} \sim p, \mathbf{a}_{t+1} \sim \rho_{\pi_{\succ}}} \big[ Q_n(\mathbf{s}_{t+1}, \mathbf{a}_{t+1}) \big] \\
&= \sum_{i=1}^{n-1} r_{\succ i}(\mathbf{s}, \mathbf{a}) + Q_n(\mathbf{s}, \mathbf{a}) = \sum_{i=1}^{n-1} \ln c_i(\mathbf{s}, \mathbf{a}) + Q_n(\mathbf{s}, \mathbf{a}).
\end{aligned}
\tag{18}
$$

This shows that we only have to learn the $n$-th subtask Q-function for $r_{\succ}(\mathbf{s}, \mathbf{a})$ during the adaptation step, because the transformed Q-values of all $n-1$ higher-priority subtask Q-functions are know by construction.

Lastly, since we are using the MaxEnt framework we have the proportional relationship

$$
\begin{aligned}
\pi_{\succ}(\mathbf{a} \mid \mathbf{s}) &\propto \exp\left( Q_{\succ}(\mathbf{s}, \mathbf{a}) \right) \\
&= \exp\left( \sum_{i=1}^{n-1} \ln(c_i(\mathbf{s}, \mathbf{a})) + Q_n(\mathbf{s}, \mathbf{a}) \right) \\
&= \left( \prod_{i=1}^{n-1} c_i(\mathbf{s}, \mathbf{a}) \right) \exp Q_n(\mathbf{s}, \mathbf{a})
\end{aligned}
\tag{19}
$$

and can see that

$$
\pi_{\succ}(\mathbf{a} \mid \mathbf{s}) \propto \begin{cases} \exp Q_n(\mathbf{s}, \mathbf{a}) & \text{if } \prod_{i=1}^{n-1} c_i(\mathbf{s}, \mathbf{a}) = 1, \\ 0 & \text{otherwise.} \end{cases}
\tag{20}
$$

This shows that $\pi_{\succ}$ softly maximizes $r_n(\mathbf{s}, \mathbf{a})$ in the global action indifference space $\mathcal{A}_{\succ}(\mathbf{s})$, as the policy is proportional to $\exp(Q_n(\mathbf{s}, \mathbf{a}))$ and has zero probability for constraint-violating actions.

## B  ARBITER VIEW THEORY

Here we provide an analysis of our algorithm from the perspective of the arbiter agent for the lexico-graphic MORL task. In this view, the algorithm optimizes the arbiter policy with policy evaluation and policy improvement steps in a transformed MORL problem using on-policy updates for subtask $n$, like Q-decomposition. Now we show that this learning scheme converges by considering a fixed arbiter policy $\pi_{\succ}$ with the *on-policy soft Bellman backup operator* $\mathcal{T}^{\pi_{\succ}}$ defined as

$$
\mathcal{T}^{\pi_{\succ}} Q_n(\mathbf{s}_t, \mathbf{a}_t) \triangleq r_n(\mathbf{s}_t, \mathbf{a}_t) + \gamma \mathbb{E}_{\mathbf{s}_{t+1} \sim p}[V_n^{\pi_{\succ}}(\mathbf{s}_{t+1})],
\tag{21}
$$

with

$$
V_n^{\pi_{\succ}}(\mathbf{s}_t) = \mathbb{E}_{\mathbf{a}_t \sim \pi_{\succ}}[Q_n(\mathbf{s}_t, \mathbf{a}_t) - \underbrace{\log(\pi_{\succ}(\mathbf{a}_t \mid \mathbf{s}_t))}_{\mathcal{H}}],
\tag{22}
$$

where $V_n^{\pi_{\succ}}(\mathbf{s}_t)$ is the soft, on-policy value function (Haarnoja et al., 2018b).

**Theorem B.1** (Arbiter policy evaluation). *Consider the soft Bellman backup operator $\mathcal{T}^{\pi_{\succ}}$ and an initial mapping $Q_n^0 : \mathcal{S} \times \mathcal{A} \to \mathbb{R}$ with $|\mathcal{A}| < \infty$ and define $Q_n^{l+1} = \mathcal{T}^{\pi_{\succ}} Q_n^l$. The sequence of $Q_n^l$ converges to $Q_n^{\pi_{\succ}}$, the soft Q-value of $\pi_{\succ}$, as $\to \infty$.*

*Proof.* See supplementary material Sec. F.1. $\square$

**Theorem B.2** (Arbiter policy improvement). *Given an arbiter policy $\pi_\succ$, define a new arbiter policy as $\pi'_\succ(\cdot|\mathbf{s}) \propto \exp(Q^{\pi_\succ}_\succ(\mathbf{s}, \cdot))$, $\forall \mathbf{s} \in \mathcal{S}$. Then $Q^{\pi'_\succ}_\succ(\cdot, \mathbf{s}) \geq Q^{\pi_\succ}_\succ(\cdot, \mathbf{s})$, $\forall \mathbf{s}, \mathbf{a}$.*

*Proof.* See supplementary material Sec. F.2. □

**Corollary B.3** (Arbiter policy iteration). *Let $\pi^0_\succ$ be the initial arbiter policy and define $\pi^{l+1}_\succ(\cdot \mid \mathbf{s}) \propto \exp(Q^{\pi^l_\succ}_\succ(\mathbf{s}, \cdot))$. Assume $|\mathcal{A}| < \infty$ and that $r_n$ is bounded. Then $Q^{\pi^l_\succ}_\succ$ improves monotonically and $\pi^l_\succ$ converges to $\pi^*_\succ$.*

*Proof.* See supplementary material Sec. F.3. □

## C    OFFLINE ADAPTATION

Our algorithm, PSQD, relies on the off-policy soft Bellman backup operator $\mathcal{T}$ in Eq. equation 11 and the objectives $J_Q(\theta)$ in Eq. equation 12 and $J_\pi(\phi)$ in Eq. equation 13 which can be computed without online policy rollouts. We can therefore adapt previously learned subtask solutions $Q_i, \pi_i$ offline with retained training data for subtask $i$, $(\mathbf{s}_t, \mathbf{a}_t, r_t, \mathbf{s}_{t+1}) \in \mathcal{D}_i$. However, since $\pi_i$ was unconstrained during pre-training, $\mathcal{D}_i$ likely contains constraint-violating transitions that are impossible in $\mathcal{M}_{\succ i}$. To account for this, when sampling a transition $(\mathbf{s}_t, \mathbf{a}_t, r_t, \mathbf{s}_{t+1}) \sim \mathcal{D}_i$ during adaptation of the pre-trained $Q_i, \pi_i$, we can check whether $\mathbf{a}_t \in \mathcal{A}_\succ(\mathbf{s}_t)$ and discard all constraint violating transitions. This is like sampling from a new dataset $\mathcal{D}_{\succ i}$ that contains only transitions from $\mathcal{M}_{\succ i}$. Depending on how well $\mathcal{D}_{\succ i}$ covers $\mathcal{M}_{\succ i}$, we can learn the optimal solution for the global task entirely offline, without requiring additional environment interaction. This makes our approach maximally sample-efficient for learning complex, lexicographic tasks by re-using data retained from subtask pre-training.

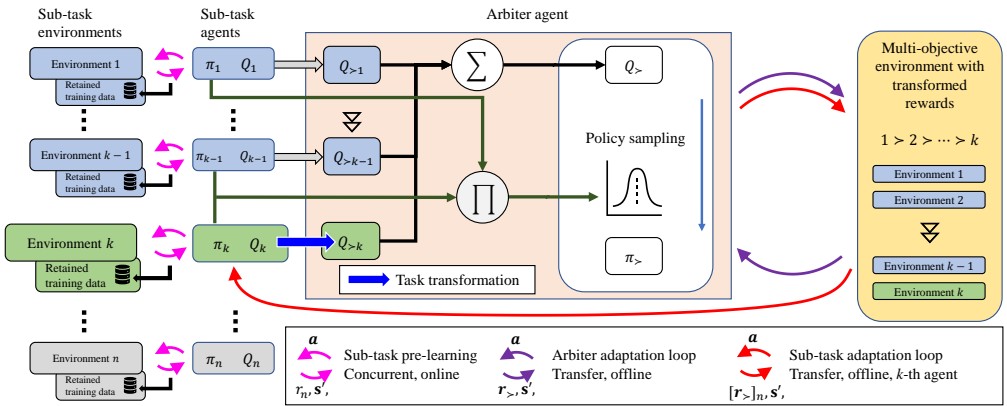

Figure 5: A high-level overview of our method. Starting on the left, $n$ agents individually learn to solve each subtask, we refer to this ⤸ as the subtask pre-training step. In the middle box, the subtask agents are combined into the lexicographic arbiter agent. The subtask adaptation loop that we implement in practice, as described in Sec. 3.2 and 3.3, is denoted by ⤸, while the arbiter learning perspective, described in App. B, is denoted by ⤸.

## D  PSQD ALGORITHM DETAILS

Here, we provide additional details on our method. A high-level overview of our framework and learning algorithm is given in Fig. 5. As can be seen, we begin by pre-training agents on the un-transformed, original subtasks $r_1, \ldots, r_n$. During the pre-training step, the agents are learning in isolation and greedily optimize their respective reward signals. Different learning algorithms could be used for pre-training the subtask agents, but in our case, the pre-training step corresponds to running $n$ instance of Soft Q-Learning Haarnoja et al. (2017). The pseudo-code for pre-training is given in Alg. 1.

---

**Algorithm 1** Subtask pre-training with SQL

---

**Require:** Subtask reward functions $r_1, \ldots, r_n$
    Samplers $\leftarrow \{\}$
    Q-functions$\leftarrow \{\}$
    Buffers $\leftarrow \{\}$
    **for** $i \in \{1, \ldots, n\}$ **do**                      ▷ Iterate (concurrently) over subtasks $\{1, \ldots, n\}$
        Initialize $\phi_i, \theta_i, \bar{\theta}_i, \mathcal{D}_i$           ▷ Initialize subtask network parameters and empty buffer
        **for** $t \in \{1, \ldots, T\}$ **do**                            ▷ $T$ total steps
            **Online interaction**
            $\zeta_t \sim P$                    ▷ Random noise for ASVGD sampling network
            $\mathbf{a}_t \sim f_i^{\phi_i}(\zeta_t, \mathbf{s}_t)$       ▷ **Unconstrained** action selection using sampling network
            $\mathbf{s}_{t+1} \sim p(\mathbf{s}_{t+1} \mid \mathbf{s}_t, \mathbf{a}_t)$
            $\mathcal{D} \cup \mathcal{D} \leftarrow \{\mathbf{s}_t, \mathbf{a}_t, r_i(\mathbf{s}_t, \mathbf{a}_t), \mathbf{s}_{t+1}\}$       ▷ Store transition and **subtask** reward
            **Update networks**
            Update $\theta_i$                       ▷ SQL Q-function update
            Update $\phi_i$                    ▷ SQL sampling network update
            $\bar{\theta}_i \leftarrow \tau\theta_i + (1 - \tau)\bar{\theta}_i$           ▷ Polyak target network update
        **end for**
        Samplers $\leftarrow$ Samplers $\cup\{\phi_i\}$
        Q-functions $\leftarrow$ Q-functions $\cup\{\theta_i\}$
        Buffers $\leftarrow$ Buffers $\cup\{\mathcal{D}_i\}$
    **end for**
    **return** Subtask sampling networks $\{\phi_1, \ldots, \phi_n\}$, Q-functions $\{\theta_1 \ldots \theta_n\}$, and Buffers $\{\mathcal{D}_1, \ldots, \mathcal{D}_n\}$

---

After the SQL pre-training step in Alg 1, we have access to converged subtask Q-functions, sampling networks for those Q-functions, and populated replay buffers for each subtask $r_i$. Next, we perform our adaption step where we finetune the pre-trained Q-functions and sampling networks under the consideration of the lexicographic constraint on higher-priority subtask performance. This corresponds to the subtask adaptation loop in Figure 5. Importantly, adapting the $i$-th subtask Q-function and sampling network requires that all higher-priority subtasks components have already been adapted to the lexicographic setting because otherwise, their indifference spaces might not overlap (see also App. E). We therefore employ an iterative adaptation procedure where we start with the second highest task (the highest-priority task is unconstrained) and adapt it under consideration of the lexicographic constraint for the highest-priority task. Then, the third-highest-priority task can be adapted under consideration of the lexicographic constraint for the highest- and second-highest-priority tasks, and so on.

---

**Algorithm 2** Incremental PSQD subtask adaptation

---

**Require:** Subtask reward functions $r_1, \ldots, r_n$
**Require:** Threshold scalars $\varepsilon_1, \ldots, \varepsilon_{n-1}$
**Require:** Pre-trained subtask sampling network parameters $\phi_1, \ldots, \phi_n$
**Require:** Pre-trained subtask Q-function parameters $\theta_1, \ldots, \theta_n$
**Require:** Populated subtask replay buffers $\mathcal{D}_1, \ldots, \mathcal{D}_n$

    **for** $i \in \{2, \ldots, n\}$ **do**          $\triangleright$ Iterate over subtasks, starting with second-highest priority task
         $\mathcal{D} = \mathcal{D} - \{\mathbf{s}_t, \mathbf{a}_t, r_i(\mathbf{s}_t, \mathbf{a}_t), \mathbf{s}_{t+1}\} \forall \mathbf{a}_t \notin \mathcal{A}_\succ$      $\triangleright$ Discard **constraint-violating** transitions
         **for** $t \in \{1, \ldots, T\}$ **do**                    $\triangleright$ $T$ total steps

             **Optional online interaction**
             // As described in Sec. 3.3, we perform importance sampling to sample from $\pi_{\succ i}$, the
             // lexicographically constrained policy for the current subtask. The already adapted
             // $\phi_1, \ldots, \phi_{i-1}$ and the current $\phi_i$ sampling networks are used as proposal distributions,
             // with the unnormalized target density being given by Eq. 10.
             $\mathbf{a}_t \sim \pi_{\succ i}$
             $\mathbf{s}_{t+1} \sim p(\mathbf{s}_{t+1} \mid \mathbf{s}_t, \mathbf{a}_t)$
             $\mathcal{D} \cup \mathcal{D} \leftarrow \{\mathbf{s}_t, \mathbf{a}_t, r_i(\mathbf{s}_t, \mathbf{a}_t), \mathbf{s}_{t+1}\}$          $\triangleright$ Store transition and **subtask** reward

             **Update networks**
             Update $\theta_i$ using Eq. 12
             Update $\phi_i$ using Eq. 13
             $\bar{\theta}_i \leftarrow \tau\theta_i + (1-\tau)\bar{\theta}_i$                  $\triangleright$ Polyak target network update
         **end for**
     **end for**
     **return** Adapted sampling network parameters $\phi_i$ and Q-function parameters $\theta_i$

---

Notice that the online adaptation step in Alg. 2 is optional, it is also possible to compute the network updates entirely offline, using the filtered versions of the subtask replay buffers from pre-training. The filtered versions of the replay buffer are obtained as described in App. C, where we ensure that no constrain-violating transitions are used during offline adaptation. If additional online interaction data should be collected from the environment, this can be done by sampling actions from the constrained policy for the current subtask.

# E    DISCUSSION ON SEMANTICALLY INCOMPATIBLE SUBTASKS

Here, we discuss an interesting edge case, namely, situations where subtasks are semantically incompatible. Let us consider an example where we wish to jointly optimize three subtasks: Subtask $r_1$ rewards moving to the left, $r_2 = -r_1$ rewards moving to the right, and $r_3$ rewards moving to the top. Clearly, $r_1$ and $r_2$ are semantically incompatible, since $r_2$ is the inverse of $r_1$. We should first notice that such MORL tasks with semantically incompatible subtasks are inherently ill-posed, since no agent can, at the same time, behave optimally for both $r_1$ and $r_2$. Intuitively, an agent can either move to the left, or to the right, but it can not tend to both directions at the same time. We emphasize the *semantic* incompatibility, since, from the MORL algorithm's perspective, tasks can not be "incompatible". Even when $r_2 = -r_1$, maximizing the summed reward $r_1 + r_2$ is still valid, even though the optimal behavior for this task might not be aligned with the RL practitioner hoped to obtain.

In the general case, given semantically incompatible subtasks, a non-lexicographic MORL agent that maximizes $r_1 + r_2 + r_3$ will likely fail to progress on either of the incompatible subtasks, as shown by (Russell & Zimdars, 2003). The behavior depends on subtask reward scale, the discount factor, and function approximation errors, which can be considerable since deep regression for Bellman optimality tends to overestimate state-action values Mnih et al. (2015). Therefore, MORL algorithms without subtask priorities are likely to produce unexpected and undesired behavior when subtasks are semantically incompatible.

Importantly, if we instead optimize the lexicographic task $r_1 \succ r_2 \succ r_3$, most of the above issues are alleviated: The lexicographic problem definition states clearly how the agent ought to behave, since, no matter how different the subtasks might be, near-optimal performance of high-priority tasks is required by definition. For the same reason, lexicographic tasks are also insensitive with respect to subtask reward scale and discount factors, since they are not optimizing a weighted combination of subtask rewards. In fact, lexicographic RL was introduced and motivated by (Gábor et al., 1998) to precisely resolve situations with semantically incompatible subtasks, e.g. "Buridan's Ass" dilemma, which is a philosophical paradox that translates directly to conflicting subtasks in MORL.

While lexicographic RL inherently resolves subtask incompatibility via its problem definition, we now clarify how our learning algorithm resolves these situations in practice. First, we notice that greedy subtask solutions obtained through SQL pre-training in Alg. 1 can indeed be "incompatible". Pre-training independently on our exemplary tasks $r_1$ and $r_2 = -r_1$ would result in Q-functions $Q_1^*$ and $Q_2^*$ that assign high-value to inverse parts of the action space (moving left and moving right). Thus, the intersection of the indifference spaces given by $Q_1^*$ and $Q_2^*$ can be empty, which would imply an empty action space for lower-priority subtasks and an ill-posed learning problem.

Our definition of the lexicographic constraint in Eq. 7, however, formulates the lexicographic constraints on the on-policy Q-functions for the constraint-respecting arbiter agent. Our learning and adaption procedure in Alg. 2 accounts for this since it *sequentially* adapts subtask solutions, starting with higher-priority subtasks. This means that in the first iteration, we adapt the solution of the subtask with second-highest priority to the lexicographic optimality constraint on the highest-priority subtasks. This changes the greedy subtask agent (i.e. its Q-function) $Q_2^* \rightsquigarrow Q_{\succ 2}^*$ to an agent that is no longer greedy with respect to its original subtask – the adapted agent instead solves $r_2$ as well as possible, while respecting the lexicographic constraint. As described in App. A, $Q_{\succ 2}^*$ assigns $-\infty$ to all actions that violate the lexicographic constraint, therefore, the high-value region and indifference space of $Q_{\succ 2}^*$ has to be inside the high-value region in $Q_1^*$, i.e. inside the indifference space of the higher-priority task $r_1$. Thus, although $Q_1^*$ and $Q_2^*$ might be "incompatible", the adapted $Q_{\succ 2}^*$ is indeed compatible with $Q_1^*$, even when $r_2 = -r_1$. Since in our adaption procedure, lower-priority subtasks always use the already adapted subtask solutions for higher-priority tasks, the intersection of all higher-priority indifference spaces can never be empty.

In summary, lexicographic MORL tasks resolve incompatible subtasks by definition, since lower-priority subtasks are constrained to solutions that are also near-optimal for all higher-priority subtasks. Our adaption procedure accounts for this by adapting subtasks sequentially, which ensures that the intersection of all higher-priority subtasks is never empty.

# F PROOFS

In Sec. F.1 to Sec. F.3 we prove the "on-arbiter-policy" view of our algorithm. We prove the subtask view of our algorithm in Sec. F.4. These proofs are based on and make use of the following observation. Due to the incremental nature of our learning algorithm, the $n-1$ higher-priority tasks have already been learned in an on-arbiter-policy fashion and are fixed during learning of the $n$ subtask Q-function. Thus we have access to the transformed subtask Q-functions $Q_{\succ i}$ for all higher priority tasks $1, \ldots, n-1$, which are stationary during learning of the $n$-th subtask Q-function. As a consequence, the arbiter policy $\pi_{\succ}$, which we want to improve and for which we want to learn $n$-th subtask Q-function in an on-policy fashion, is already constrained to the global indifference space and can never select constraint-violating actions. Thus, we can rely on existing proofs for MaxEnt RL, since our arbiter policy is just a particular MaxEnt policy.

## F.1 ARBITER POLICY EVALUATION

**Lemma F.1.** *Arbiter policy evaluation. Consider the soft Bellman backup operator $\mathcal{T}^{\pi}$ and an initial mapping $Q_n^0 : \mathcal{S} \times \mathcal{A} \to \mathbb{R}$ with $|\mathcal{A}| < \infty$ and define $Q_n^{l+1} = \mathcal{T}^{\pi} Q_n^l$. The sequence of $Q_n^l$ will converge to $Q_n^{\pi}$, the soft Q-value of $\pi$, as $l \to \infty$.*

The proof is straightforward based on the observation that the $n$-th subtask Q-function is simply a soft Q-function for the task $r_n$, learned under the expectation of the global arbiter policy (Russell & Zimdars, 2003), which has zero-probability for constraint-violating actions. Thus, the proof is analogous to the one by Haarnoja et al. (2018b) except in our case $\pi$ is a constraint-respecting arbiter policy. We repeat the proof here only for completeness:

*Proof.* We define the on-policy, entropy augmented reward signal for soft Q-functions as $r_n^{\pi}(\mathbf{s}_t, \mathbf{a}_t) \triangleq r_n(\mathbf{s}_t, \mathbf{a}_t) + \mathbb{E}_{\mathbf{s}_{t+1} \sim p}[\mathcal{H}(\pi_{\succ}(\cdot|\mathbf{s}_{t+1}))]$ and rewrite $\mathcal{T}^{\pi}$ as

$$Q_n^{\pi}(\mathbf{s}_t, \mathbf{a}_t) \leftarrow r_n^{\pi}(\mathbf{s}_t, \mathbf{a}_t) + \gamma \mathbb{E}_{\mathbf{s}_{t+1} \sim p, \mathbf{a}_{t+1} \sim \pi}[Q_n^{\pi}(\mathbf{s}_{t+1}, \mathbf{a}_{t+1})]. \tag{23}$$

The standard convergence result for policy evaluation (Sutton & Barto, 2018) thus holds. The assumption $|\mathcal{A}| < \infty$ is required to guarantee that the entropy augmented reward is bounded. $\square$

## F.2 ARBITER POLICY IMPROVEMENT

We want to show that we can obtain an arbiter policy $\pi_{\succ}$ that is better or as least as good as any other arbiter policy by softly maximizing the soft global Q-function. Since the arbiter policy maximizes the sum of all (transformed) subtask Q-functions we set $r = r_{\succ} = \sum_{j=1}^{n-1} r_{\succ j} + r_n$ and $Q_{\text{soft}}^{\pi} = Q_{\succ}^{\pi} = \sum_{j=1}^{n-1} Q_{\succ j} + Q_n$ for the remainder of this section.

Since we are concerned with finding an improved arbiter policy and since all arbiter policies have zero probability for selecting constraint-violating actions, the $r_{\succ j}$ and $Q_{\succ j}$ terms evaluate to zero and we can only improve $Q_{\text{soft}}^{\pi}$ by increasing $r_n$. $Q_{\text{soft}}^{\pi}$ still correctly assigns a value of $-\infty$ to actions that violate constraints, however, this value is known through the constraint on already optimal higher-priority Q-functions and not backed up while learning task $r_n$. In practice, since arbiter policies can never select constraint-violating actions, the soft policy improvement theorem by Haarnoja et al. (2017) also holds for us since we are softly maximizing the ordinary task $r_n$ with a particular soft policy. Arbiter policy improvement for the global tasks thus degrades to maximizing the $n$-th task while respecting priority constraints. In the following, we use the short-hand $r_t = r(\mathbf{s}_t, \mathbf{a}_t)$ for a reward at time $t$ for more compact notation.

**Theorem F.2.** *Arbiter policy improvement. Given an arbiter policy $\pi$, define a new arbiter policy as $\pi'(\cdot|\mathbf{s}) \propto \exp(Q_{\succ}^{\pi}(\mathbf{s}, \cdot))$, $\forall \mathbf{s} \in \mathcal{S}$. Then $\pi'(\cdot|\mathbf{s}) \geq \pi(\cdot|\mathbf{s})$, $\forall \mathbf{s}, \mathbf{a}$.*

*Proof.* The soft policy maximization objective

$$J(\pi) \triangleq \sum_t \mathbb{E}_{(\mathbf{s}_t, \mathbf{a}_t) \sim \rho_{\pi}}[Q_{\text{soft}}^{\pi}(\mathbf{s}_t, \mathbf{a}_t) + \mathcal{H}(\pi(\cdot|\mathbf{s}_t))] \tag{24}$$

can be improved either by increasing the soft Q-value or the entropy term:

$$\mathbb{E}_{(\mathbf{s}_t, \mathbf{a}_t) \sim \rho_{\pi}}[Q_{\text{soft}}^{\pi}(\mathbf{s}_t, \mathbf{a}_t)] + \mathcal{H}(\pi(\cdot|\mathbf{s}_t)) \leq \mathbb{E}_{(\mathbf{s}_t, \mathbf{a}_t) \sim \rho_{\pi}}[Q_{\text{soft}}^{\pi}(\mathbf{s}_t, \mathbf{a}_t)] + \mathcal{H}(\pi'(\cdot|\mathbf{s}_t)). \tag{25}$$

Recalling the definition of the soft Q-function

$$Q_{\text{soft}}^\pi(\mathbf{s}_t, \mathbf{a}_t) \triangleq r_0(\mathbf{s}_t, \mathbf{a}_t) + \mathbb{E}_{(\mathbf{s}_{t+1}, \mathbf{a}_{t+1}, \dots) \sim \rho_\pi}\left[\sum_{l=1}^\infty \gamma^l(r(\mathbf{s}_{t+l}, \mathbf{a}_{t+l}) + \mathcal{H}(\mathbf{a}_{t+l} \mid \mathbf{s}_{t+l}))\right], \quad (26)$$

the proof for the (soft) policy improvement theorem now simply expands the right side of equation 25 with one-step look-aheads until one obtains $Q_{\text{soft}}^{\pi'}$:

$$
\begin{aligned}
Q_{\text{soft}}^\pi(\mathbf{s}_0, \mathbf{a}_0) &= r_0 + \mathbb{E}_{\mathbf{s}_1}\Big[\gamma\big(\mathcal{H}(\pi(\cdot|\mathbf{s}_1)) + \mathbb{E}_{\mathbf{a}_1 \sim \pi}[Q_{\text{soft}}^\pi(\mathbf{s}_1, \mathbf{a}_1)]\big)\Big] \\
&\leq r_0 + \mathbb{E}_{\mathbf{s}_1}\Big[\gamma\big(\mathcal{H}(\pi'(\cdot|\mathbf{s}_1)) + \mathbb{E}_{\mathbf{a}_1 \sim \pi'}[Q_{\text{soft}}^\pi(\mathbf{s}_1, \mathbf{a}_1)]\big)\Big] \\
&= r_0 + \mathbb{E}_{\mathbf{s}_1}\Big[\gamma\big(\mathcal{H}(\pi'(\cdot|\mathbf{s}_1)) + r_1\big)\Big] + \gamma^2 \mathbb{E}_{\mathbf{s}_2}\Big[\mathcal{H}(\pi(\cdot|\mathbf{s}_2)) + \mathbb{E}_{\mathbf{a}_2 \sim \pi}[Q_{\text{soft}}^\pi(\mathbf{s}_2, \mathbf{a}_2)]\Big] \\
&\leq r_0 + \mathbb{E}_{\mathbf{s}_1}\Big[\gamma\big(\mathcal{H}(\pi'(\cdot|\mathbf{s}_1)) + r_1\big)\Big] + \gamma^2 \mathbb{E}_{\mathbf{s}_2}\Big[\mathcal{H}(\pi'(\cdot|\mathbf{s}_2)) + \mathbb{E}_{\mathbf{a}_2 \sim \pi'}[Q_{\text{soft}}^\pi(\mathbf{s}_2, \mathbf{a}_2)]\Big] \\
&= r_0 + \mathbb{E}_{(\mathbf{s}_1, \mathbf{s}_2) \sim p, (\mathbf{a}_1, \mathbf{a}_2) \sim \pi'}\Big[\gamma\big(\mathcal{H}(\pi'(\cdot|\mathbf{s}_1)) + r_1\big)\Big] + \gamma^2 \mathcal{H}(\pi'(\cdot|\mathbf{s}_2)) + Q_{\text{soft}}^\pi(\mathbf{s}_2, \mathbf{a}_2) \\
&\vdots \\
&\leq r_0 + \mathbb{E}_{(\mathbf{s}_{t+1}, \mathbf{a}_{t+1}, \dots) \sim \rho_{\pi'}}\left[\sum_{l=1}^\infty \gamma^l(r_{t+l} + \mathcal{H}(\pi'(\cdot \mid \mathbf{s}_{t+l})))\right] \\
&= Q_{\text{soft}}^{\pi'}(\mathbf{s}_0, \mathbf{a}_0)
\end{aligned}
$$

$$(27)$$

$\square$

### F.3 ARBITER POLICY ITERATION

**Corollary F.3** (Arbiter policy iteration). *Let $\pi_\succ^0$ be the initial arbiter policy and define $\pi_\succ^{l+1}(\cdot \mid \mathbf{s}) \propto \exp(Q_\succ^{\pi_\succ^l}(\mathbf{s}, \cdot))$. Assume $|\mathcal{A}| < \infty$ and that $r_n$ is bounded. Then $Q_\succ^{\pi_\succ^l}$ improves monotonically and $\pi_\succ^l$ converges to $\pi_\succ^*$.*

*Proof.* Let $\pi_l$ be the arbiter policy at iteration $l$. According to Theorem B.2, the sequence $Q_\succ^{\pi_\succ^l}$ improves monotonically. Since arbiter policies only select actions inside the global action indifference space, the $\ln(c_i)$ components of $Q_\succ$ evaluate to zero and since $r_n$ is assumed to be bounded and since the entropy term is bounded by the $|\mathcal{A}| < \infty$ assumption, the sequences converges to some $\pi_\succ^*$. $\pi_\succ^*$ is optimal, since according to Theorem B.2 we must have $Q_\succ^{\pi_\succ^*} > Q_\succ^{\pi_\succ}, \forall \pi_\succ \neq \pi_\succ^*$. $\square$

### F.4 PRIORITIZED SOFT Q-LEARNING

We now prove the local view on our algorithm, by showing that we can define an off-policy backup operator that accurately reflects the global arbiter policy, thus connecting the global on-policy and local off-policy views on our algorithm. For this, we first note the soft Bellman optimality equations that have already been proven by Haarnoja et al. (2017)

$$Q_{\text{soft}}^*(\mathbf{s}_t, \mathbf{a}_t) = r(\mathbf{s}_t, \mathbf{a}_t) + \gamma \mathbb{E}_{\mathbf{s}_{t+1} \sim p}[V_{\text{soft}}^*(\mathbf{s}_{t+1})], \quad (28)$$

with

$$V_{\text{soft}}^*(\mathbf{s}_t) = \underbrace{\log \int_\mathcal{A} \exp\left(\frac{1}{\alpha} Q_{\text{soft}}^*(\mathbf{s}_t, \mathbf{a}')\right) d\mathbf{a}'}_{\text{softmax}}, \quad (29)$$

where the softmax operator represents a policy $\pi_{\text{softmax}}$ that softly maximizes $Q^*$. Thus, we can also write

$$Q_{\text{soft}}^*(\mathbf{s}_t, \mathbf{a}_t) = r(\mathbf{s}_t, \mathbf{a}_t) + \gamma \mathbb{E}_{\mathbf{s}_{t+1} \sim p, \mathbf{a}_{t+1} \sim \pi_{\text{softmax}}}[Q_{\text{soft}}^*(\mathbf{s}_{t+1}, \mathbf{a}_{t+1})], \quad (30)$$

which shows that soft Q-learning is equivalent to Q-learning with softmax instead of greedy max action selection.

We now show that we can learn the optimal $n$-th subtask Q-function under the arbiter policy in an off-policy, soft Q-learning fashion. In principal, this simply replaces the softmax policy $\pi_{\text{softmax}}$ with the the arbiter policy $\pi_{\succ}$ in equation 30. However, the arbiter policy does not softly maximize the $n$-th subtask Q-function like the locally greedy softmax policy in equation 29 would do. Instead, it softly maximizes the sum of subtask Q-functions $Q_{\succ}(\mathbf{s}, \mathbf{a}) = \sum_{j=1}^{n-1} Q_{\succ j}(\mathbf{s}, \mathbf{a}) + Q_n(\mathbf{s}, \mathbf{a})$. To show that we can approximate the global arbiter oplicy in an off-policy fashion, we split the action space of the MDP into two disjoints sub-spaces $\mathcal{A} = \mathcal{A}_{\succ}(\mathbf{s}) \dot{\cup} \bar{\mathcal{A}}_{\succ}(\mathbf{s})$. The first sub-space is the action indifference space $\mathcal{A}_{\succ}(\mathbf{s})$ that contains all constraint respecting actions in each state, while the second sub-space $\bar{\mathcal{A}}_{\succ}(\mathbf{s})$ contains all constraint violating actions. Thus, we can split the integral in equation 29

$$V_{\text{soft}}^*(\mathbf{s}_t) = \log \int_{\mathcal{A}_{\succ}(\mathbf{s}_t)} \exp\left(Q_{\text{soft}}^*(\mathbf{s}_t, \mathbf{a}_{\succ})\right) d\mathbf{a}_{\succ} + \log \int_{\bar{\mathcal{A}}_{\succ}(\mathbf{s}_t)} \exp\left(Q_{\text{soft}}^*(\mathbf{s}_t, \bar{\mathbf{a}})\right) d\bar{\mathbf{a}}. \tag{31}$$

We now insert our optimal global Q-function and obtain

$$V_{\succ}^*(\mathbf{s}_t) = \log \int_{\mathcal{A}_{\succ}(\mathbf{s})} \exp\left(Q_{\succ}^*(\mathbf{s}_t, \mathbf{a}_{\succ})\right) d\mathbf{a}_{\succ} + \log \int_{\bar{\mathcal{A}}_{\succ}(\mathbf{s})} \exp\left(Q_{\succ}^*(\mathbf{s}_t, \bar{\mathbf{a}})\right) d\bar{\mathbf{a}}, \tag{32}$$

which immediately simplifies to

$$\begin{aligned} V_{\succ}^*(\mathbf{s}_t) &= \log \int_{\mathcal{A}_{\succ}(\mathbf{s})} \exp\left(Q_{\succ}^*(\mathbf{s}_t, \mathbf{a}_{\succ})\right) d\mathbf{a}_{\succ} \\ &= \log \int_{\mathcal{A}_{\succ}(\mathbf{s})} \exp\left(Q_n^*(\mathbf{s}_t, \mathbf{a}_{\succ})\right) d\mathbf{a}_{\succ}, \end{aligned} \tag{33}$$

where the second integral over $\bar{\mathcal{A}}$ disappears because actions in $\bar{\mathcal{A}}$ have zero probability under the arbiter policy while $Q_{\succ}^*$ becomes $Q_n^*$ because the $n-1$ subtask Q-function all evaluate to zero. Thus, the optimal value of a state under the arbiter policy corresponds to the softmax of the untransformed, $n$-th subtask Q-function, in the global indifference space. Thus for the optimal state-action value function of the arbiter policy we have

$$Q_{\succ}^*(\mathbf{s}_t, \mathbf{a}_t) = r(\mathbf{s}_t, \mathbf{a}_t) + \gamma \mathbb{E}_{\mathbf{s}_{t+1} \sim p}[V_{\succ}^*(\mathbf{s}_{t+1})], \tag{34}$$

which is equivalent to

$$Q_{\succ}^*(\mathbf{s}_t, \mathbf{a}_t) = r(\mathbf{s}_t, \mathbf{a}_t) + \gamma \mathbb{E}_{\mathbf{s}_{t+1} \sim p, \mathbf{a}_{t+1} \sim \pi_{\succ}}[Q_{\succ}^*(\mathbf{s}_{t+1}, \mathbf{a}_{t+1})]. \tag{35}$$

We can learn $Q_{\succ}^*$ with the off-policy, soft Bellman backup operator

$$\mathcal{T}Q(\mathbf{s}, \mathbf{a}) \triangleq r(\mathbf{s}, \mathbf{a}) + \gamma \mathbb{E}_{\mathbf{s}' \sim p}\underbrace{\left[\log \int_{\mathcal{A}_{\succ}} \exp\left(Q(\mathbf{s}_t, \mathbf{a}_{\succ})\right) d\mathbf{a}_{\succ}\right]}_{V_{\succ}(\mathbf{s}')}. \tag{36}$$

**Theorem F.4** (Prioritized soft Q-learning). *Consider the soft Bellman backup operator $\mathcal{T}$, and an initial mapping $Q^0 : \mathcal{S} \times \mathcal{A}_{\succ} \to \mathbb{R}$ with $|\mathcal{A}_{\succ}| < \infty$ and define $Q^{l+1} = \mathcal{T}Q^l$, then the sequence of $Q^l$ converges to $Q_{\succ}^*$, the soft Q-value of the optimal arbiter policy $\pi_{\succ}^*$, as $l \to \infty$.*

As we have shown, $V_{\succeq}^*$ is the value function of the global arbiter policy, while $Q_{\succ}^* = Q_n^*$ is an ordinary soft Q-function (because we are in the indifference space). Thus, the original convergence proof by Haarnoja et al. (2017) directly applies to $\mathcal{T}$, we repeat it here for completeness with some annotations:

*Proof.* We want to show that $\mathcal{T}$ is a contraction, thus we note that the definition of a contraction on some metric space $M$ with norm $d$ is

$$d(f(x), f(y)) < kd(x, y), \tag{37}$$

with $0 \leq k \leq 1$. Next we define the supremum norm for soft Q-functions as $||Q_1 - Q_2|| \triangleq \max_{\mathbf{s},\mathbf{a}} |Q_1(\mathbf{s},\mathbf{a}) - Q_2(\mathbf{s},\mathbf{a})|$ and set $\epsilon = ||Q_1 - Q_2||$. It follows that

$$
\log \int \exp(Q_1(\mathbf{s}',\mathbf{a}'))d\mathbf{a}' \leq \log \int \exp(Q_2(\mathbf{s}',\mathbf{a}') + \epsilon)d\mathbf{a}'
$$

$$
= \log \left( \exp(\epsilon) \int \exp(Q_2(\mathbf{s}',\mathbf{a}'))d\mathbf{a}' \right) \tag{38}
$$

$$
= \epsilon + \log \int \exp(Q_2(\mathbf{s}',\mathbf{a}'))d\mathbf{a}'.
$$

From the last row and using the subtraction property of inequalities, we get

$$
\log \int \exp(Q_1(\mathbf{s}',\mathbf{a}'))d\mathbf{a}' - \log \int \exp(Q_2(\mathbf{s}',\mathbf{a}'))d\mathbf{a}' \leq \epsilon \tag{39}
$$

and immediately see that the soft multi-objective Bellman operator is indeed a contraction:

$$
||\gamma \mathbb{E}_{\mathbf{s}'\sim p} \left[ \log \int_{\mathcal{A}} \exp\left(Q_1(\mathbf{s}',\mathbf{a}')\right)d\mathbf{a}' - \log \int_{\mathcal{A}} \exp\left(Q_2(\mathbf{s}',\mathbf{a}')\right)d\mathbf{a}' \right]|| \leq \gamma\epsilon = \gamma||Q_1 - Q_2|| \tag{40}
$$

Here, in the left side of the inequality, $||\mathcal{T}_\Sigma Q_1 - \mathcal{T}_\Sigma Q_2||_\Sigma$, the $r(\mathbf{s},\mathbf{a})$ terms cancel out and we can collapse the two expectations $\mathbb{E}_{\mathbf{s}'\sim p}$ into one. The same is true on the right side of the inequality. Thus $\mathcal{T}$ is a contraction with optimal Q-function as fixed point. $\square$

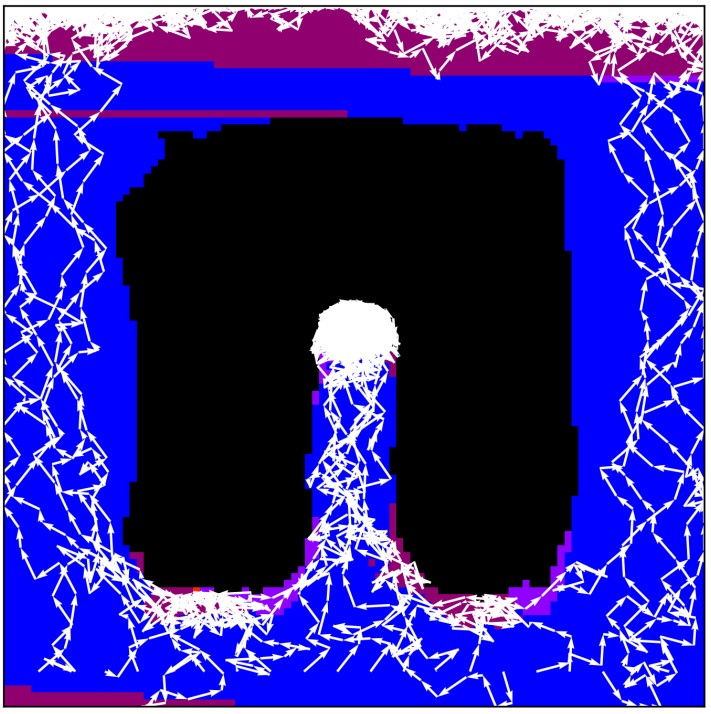

(a) Zero-shot agent for $r_{1 \succ 2}$. The agent does not collide with the obstacle but is not globally optimal, since it greedily navigates to the top and gets stuck in front of the obstacle.

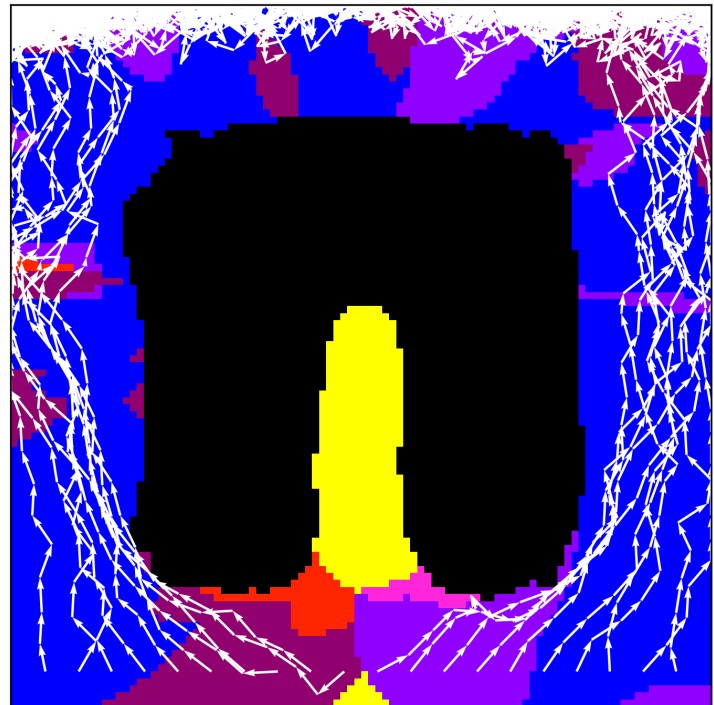

(b) Adapted agent for $r_{1 \succ 2}$. The agent has adapted to the task priority constraint and navigates around and out of the obstacle (yellow area in the middle).

$\nwarrow[-1, 1]$  $\uparrow[0, 1]$  $\nearrow[1, 1]$  $\leftarrow[-1, 0]$  $\rightarrow[1, 0]$  $\swarrow[-1, -1]$  $\downarrow[0, -1]$  $\searrow[1, -1]$

Figure 6: Larger copies of Fig. 1d and 2c. The background is colored according to discretized angles of the policy, using the colormap above.

## G EXPERIMENT DETAILS

### G.1 2D NAVIGATION ENVIRONMENT

The action space for this environment is in $\mathbb{R}^2$ and corresponds to positional changes in the $x, y$ plane, while the observation space is the current $(x, y)$ coordinate. The agent's position is clipped to the range $[-10, 10]$ for both dimensions. We normalize actions to unit length to bound the action space and penalize non-straight actions. The high-priority task $r_1$ corresponds to obstacle avoidance and yields negative rewards in close proximity to the $\cap$-shaped obstacle (see Fig. 1a)

$$r_1(\mathbf{s}) = \begin{cases} -\sigma^2 \cdot \exp(-\frac{d^2}{2 \cdot l^2}), & \text{if } d > 0 \\ -\beta - \sigma^2 \cdot \exp(-\frac{d^2}{2 \cdot l^2}) & \text{otherwise,} \end{cases} \quad (41)$$

where $d$ is obstacle distance (inferred from $\mathbf{s}$), $\sigma = 1$ and $l = 1$ parameterize a squared exponential kernel, and $\beta = 10$ is a an additional punishment for colliding with the obstacle. The auxiliary rewards $r_2$ and $r_3$ respectively yield negative rewards everywhere except in small areas at the top and at the right side of the environment

$$r_2(\mathbf{s}) = \begin{cases} 0 & \text{if } \mathbf{s}.y > 7 \\ -\delta & \text{otherwise,} \end{cases} \quad (42)$$

$$r_3(\mathbf{s}) = \begin{cases} 0 & \text{if } \mathbf{s}.x > 7 \\ -\delta & \text{otherwise,} \end{cases} \quad (43)$$

where we use $\delta = 5$ in all our experiments. Thus, the lexicographically prioritized task $r_{1 \succ 2}$ corresponds to reaching the top without colliding with the obstacle. Due to the low dimensionality of the environment we directly exploit the proportionality relationship in equation 10 and rely on importance sampling instead of learning a policy network for the 2D navigation experiments.

### G.2 FRANKA EMIKA PANDA ENVIRONMENT

This environment features a simulated Franka Emika Panda arm, shown in Fig. 4, and based on the Gymnasium Robotics package (de Lazcano et al., 2023). The action space $\mathcal{A} \in \mathbb{R}^9$ corresponds to joint positions while the state space $\mathcal{S} \in \mathbb{R}^{18}$ contains all joint positions and velocities. In this environment, the high-priority task $r_1$ corresponds to avoiding a fixed volume (red block) in the robots workspace and returns $-10$ when any number of robot joints are inside the volume. The low-priority task $r_2$ corresponds to reaching a fixed end-effector position (green sphere) and yields rewards proportional to the negative distance between the end-effector and the target plus a bonus of 100 for reaching the target. The prioritized task $r_{1 \succ 2}$ thus corresponds to reaching the target end-effector location while keeping all joints outside of the avoidance volume.

### G.3 BASELINE ALGORITHMS

Here we provide additional details on the algorithms used in the baseline comparison in Sec. 4.3. Since, to the best of our knowledge, PSQD is the first method that solves lexicographic MORL problems with *continuous* action spaces, we can not rely on existing lexicographic RL algorithms for discrete problems as baselines. Instead, we implement lexicographic task priority constraints by simplistic means in the following, state-of-the-art DRL algorithms that support continuous action spaces.

**Prioritized Soft Actor-Critic**  Soft Actor-Critic (SAC) (Haarnoja et al., 2018b;c) learns a univariate Gaussian actor by minimizing the Kullback-Leibler divergence between the policy and the normalized, soft Q-function. This policy improvement step is given by

$$\pi_{\text{new}} = \underset{\pi' \in \Pi}{\arg\min} D_{\text{KL}}\left(\pi'(\cdot, \mathbf{s}_t) \middle\| \frac{\exp\left(\frac{1}{\alpha} Q^{\pi_{\text{old}}}(\mathbf{s}_t, \cdot)\right)}{Z^{\pi_{\text{old}}}(\mathbf{s}_t)}\right), \quad (44)$$

where $\Pi$ is a set of tractable policies (e.g. parameterized Gaussians), and $Z^{\pi_{\text{old}}}(\mathbf{s}_t)$ is the normalizing constant for the unnormalized density given by $Q^{\pi_{\text{old}}}(\mathbf{s}_t, \cdot)$. To make for a lexicographic, i.e. a task priority-constrained version of SAC, a simple approach is to add a regularization term to

equation 44 to penalize lower-priority subtask policies from diverging from higher-priority subtask policies. Based on this intuition, for our prioritized SAC baseline, we augment the objective in equation 44 by adding another KL term that measures the divergence between the current subtask policy and the higher-priority subtask policy

$$\pi_{\text{new}} = \arg\min_{\pi' \in \Pi} (1 - \beta) D_{\text{KL}} \left( \pi'(\cdot, \mathbf{s}_t) \middle|\middle| \frac{\exp\left(\frac{1}{\alpha} Q^{\pi_{\text{old}}}(\mathbf{s}_t, \cdot)\right)}{Z^{\pi_{\text{old}}}(\mathbf{s}_t)} \right) + \beta D_{\text{KL}} \left( \pi'(\cdot, \mathbf{s}_t) \middle|\middle| \pi_{\text{pre}}(\cdot, \mathbf{s}_t) \right), \quad (45)$$

where $\beta$ is a weight for the convex combination of the two KL terms and $\pi_{\text{pre}}$ is the pre-trained, higher-priority policy. In practice, the parameters of the univariate Gaussian actor for our prioritized SAC are given by a DNN parameterized by $\phi$, that uses the reparametrization trick to minimize the loss

$$J_\pi(\phi) = \mathbb{E}_{\mathbf{s}_t \sim \mathcal{D}} \big[ \mathbb{E}_{\mathbf{a}_t \sim \pi_\phi} [(1 - \beta)(\alpha \log(\pi_\phi(\mathbf{a}_t \mid \mathbf{s}_t)) - Q(\mathbf{s}_t, \mathbf{a}_t)) \\ + \beta(\log \pi_\phi(\mathbf{a}_t \mid \mathbf{s}_t) - \log \pi_{\text{pre}}(\mathbf{a}_t \mid \mathbf{s}_t))]\big], \quad (46)$$

which is the original SAC objective with the additional KL regularization.

**Prioritized Proximal Policy Optimization**    To implement a prioritized version of Proximal Policy Optimization (PPO) Schulman et al. (2017), we apply the same intuition as for the prioritized SAC version, i.e. we regularize the policy to be similar to the pre-trained, higher-priority policy. PPO maximizes a clipped version of the surrogate objective

$$J_\pi(\phi) = \mathbb{E}_t \Big[ \frac{\pi_\phi(\mathbf{a}_t \mid \mathbf{s}_t)}{\pi_{\phi_{\text{old}}}(\mathbf{a}_t \mid \mathbf{s}_t)} \hat{A}_t \Big], \quad (47)$$

where $\hat{A}_t$ is an estimator of the advantage function for task return at timestep $t$ and $\phi_{\text{old}}$ is the parameter vector of the policy before doing an update. To encourage learning a policy that respects task priority constraints, we change the maximization to

$$J_\pi(\phi) = \mathbb{E}_t \left[ \frac{\pi_\phi(\mathbf{a}_t \mid \mathbf{s}_t)}{\pi_{\phi_{\text{old}}}(\mathbf{a}_t \mid \mathbf{s}_t)} \left( (1 - \beta)\hat{A}_t - \beta D_{\text{KL}} \left( \pi'(\cdot, \mathbf{s}_t) \middle|\middle| \pi_{\text{pre}}(\cdot, \mathbf{s}_t) \right) \right) \right], \quad (48)$$

such that the update now increases the probability of actions that have positive advantage and similar probability under the current and pre-trained, higher-priority policy.

**PSQD Ablation**    For the ablation to PSQD, we also pre-train the higher-priority Q-function for the obstacle avoidance task and rely on a the $c_1(\mathbf{s}, \mathbf{a})$ priority constraint indicator functions. However, instead of using them to project the policy into the indifference space, as PSQD does, we use them as an indicator for a punishment of -100 that is subtracted from the lower-priority task reward whenever the agent selects a constraint-violating action.

**Soft Q-Decomposition**    With "Soft Q-Decomposition" we refer to a continuous version of Russell & Zimdars (2003)'s discrete Q-Decomposition algorithm that we describe in Sec. 2.1. This algorithm concurrently learns the subtask Q-functions $Q_i$ for a MORL task with vectorized reward function, such that the overall Q-function, $Q_\Sigma$, can be recovered from the sum of subtask Q-functions. A key property of this algorithm is that the constituent Q-functions are learned on-policy for a central arbiter agent that maximizes the sum of rewards. To adapt the discrete Q-Decomposition algorithm to continuous action spaces, we combine it with the Soft Q-Learning (SQL) (Haarnoja et al., 2017) algorithm. However, SQL is a soft version of Q-learning (Watkins & Dayan, 1992) and thereby off-policy, thus we can not directly use SQL to learn the constituent Q-functions. This is because the SQL update to the Q-function

$$J_Q(\theta) = \mathbb{E}_{\mathbf{s}_t, \mathbf{a}_t \sim \mathcal{D}} \left[ \frac{1}{2} \left( Q_n^\theta(\mathbf{s}_t, \mathbf{a}_t) - r_n(\mathbf{s}_t, \mathbf{a}_t) + \gamma \mathbb{E}_{\mathbf{s}_{t+1} \sim p} \left[ V_n^{\bar{\theta}}(\mathbf{s}_{t+1}) \right] \right)^2 \right] \quad (49)$$

$$V_n^{\bar{\theta}}(\mathbf{s}_t) = \log \int_{\mathcal{A}} \exp(Q_n^{\bar{\theta}}(\mathbf{s}_t, \mathbf{a}')) \, d\mathbf{a}', \quad (50)$$

calculates the value of the next state by integrating over the entire action space, thereby finding the softmax of the next state value. In practice, this integral is not computed exactly, but instead approximated with actions sampled from some proposal distribution $q_{\mathbf{a}'}$

$$V_n^{\bar{\theta}}(\mathbf{s}_t) = \log \mathbb{E}_{q_{\mathbf{a}'}} \left[ \frac{\exp \frac{1}{\alpha} Q^{\bar{\theta}}(\mathbf{s}_t, \mathbf{a}_t)}{q_{\mathbf{a}'}(\mathbf{a}')} \right], \quad (51)$$

which typically is the current policy. Thus, to fix the illusion of control in SQL update to the subtask Q-functions, it suffices to use the arbiter policy instead of the greedy subtask policy for $q_{\mathbf{a}'}$. This way, the approximate soft value function in equation 51 assigns state values that correspond to the real softmax ("LogSumExp" expression) of Q-values for actions sampled from the arbiter policy.

## H  ADDITIONAL RESULTS

Here we provide additional qualitative results from the 2D navigation environment that aim to make for additional intuition on our method and how it solves lexicographic MORL problems.

### H.1  OBSTACLE INDIFFERENCE SPACE

To provide further intuition for how our composed agent implements constraints, we provide additional examples for $Q_\succ$. In Fig. 7, we plot $Q_\succ = \ln(c_1) + \hat{Q}_2^{\pi_\succ}$ at multiple locations in the environment. The black areas indicate $\bar{\mathcal{A}}_\succ$, the areas forbidden by the higher-priority obstacle avoidance task, which remain fixed during subsequent adaptation steps. The non-black areas correspond to $\hat{Q}_2^{\pi_\succ}$ in $\mathcal{A}_\succ$, where the lower-priority tasks can learn and express its preferences.

### H.2  MULTIPLE CONSTRAINTS

We illustrate tasks with multiple lexicographic priority constraints in the 2D environment. For this, we consider the prioritized tasks $r_{1\succ2\succ3}$ and $r_{1\succ3\succ2}$. Both of these lexicographic MORL tasks assign the highest priority to the obstacle avoidance subtask $r_1$, but prioritize $r_2$ (moving to the top) and $r_3$ (moving to the right) differently. The prioritized task $r_{1\succ2\succ3}$ assigns higher priority to reaching the top than reaching the side, while $r_{1\succ3\succ2}$ assigns higher priority to reaching the side than reaching the top. This is a concrete example of how task priorities can be used to model different, complex tasks. The composed, adapted agents corresponding to these lexicographic tasks are visualized in Fig. 8a and 8b. Each white arrow corresponds to one action taken by the agent.

As can be seen, the behavior of the agents for the different prioritized tasks differs noticeably, which is expected for different task prioritizations. The learned agent for $r_{1\succ2\succ3}$ first drives to the top, then to the side, while the agent for $r_{1\succ3\succ2}$ first drives to the side, then to the top. We can again explain this behavior by inspecting the components of these agents in Fig. 8c and 8b. For this, we place the agent at marker 5 in Fig. 7e and plot the constraint indicator functions for the higher priority tasks as well as the Q-function for the low priority task, for the entire action space. Both agents are constrained by $\ln(c_1)$ which induces $\bar{\mathcal{A}}_{\succ1}$ and forbids actions that bring the agent into close proximity of the obstacle. $\ln(c_2)$ in Fig. 8c induces $\bar{\mathcal{A}}_{\succ2}$, which constraints $\pi_\succ$ to actions that move up. $\ln(c_3)$ in Fig. 8d, on the other hand, restricts the $\pi_\succ$ to actions that move to the side.

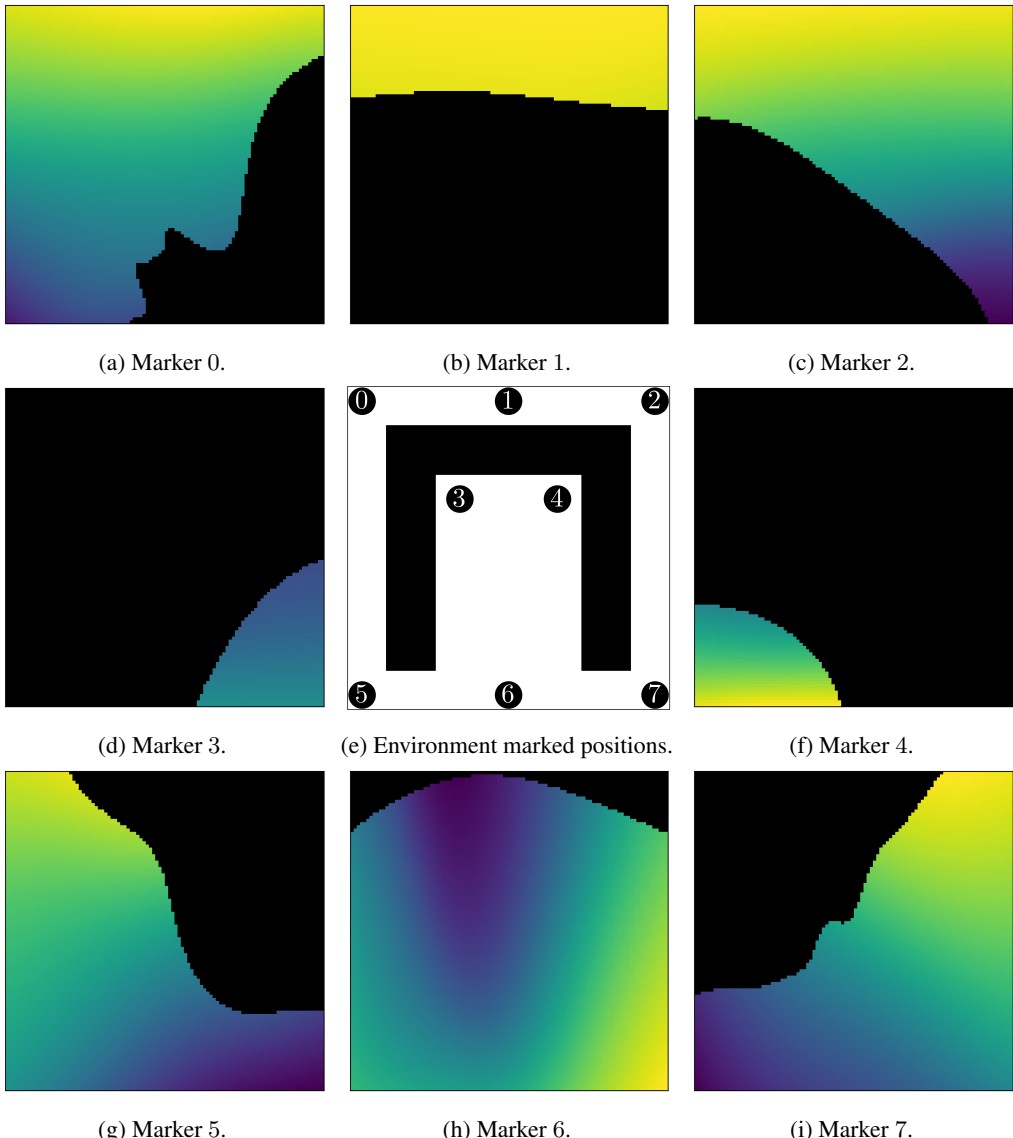

(a) Marker 0.

(b) Marker 1.

(c) Marker 2.

(d) Marker 3.

(e) Environment marked positions.

(f) Marker 4.

(g) Marker 5.

(h) Marker 6.

(i) Marker 7.

Figure 7: Analysis of the global Q-function in action space: $Q_\succ = \ln(c_1) + \hat{Q}_2^{\pi_\succ}$ at different positions in the 2D environment. The $x$ and $y$ axes of the images correspond to the $x$ and $y$ components of the action space. Brighter colored (yellow) hues indicate high probability, while darker colored (blue) hues indicate low probability under $\pi_\succ$. The black mask shows $\bar{\mathcal{A}}_\succ$. As can be seen, the task priority constraint forbids actions that lead to obstacle collisions, since those actions are not within the $\varepsilon_1$ value range of the optimal action for the high-priority task 1.

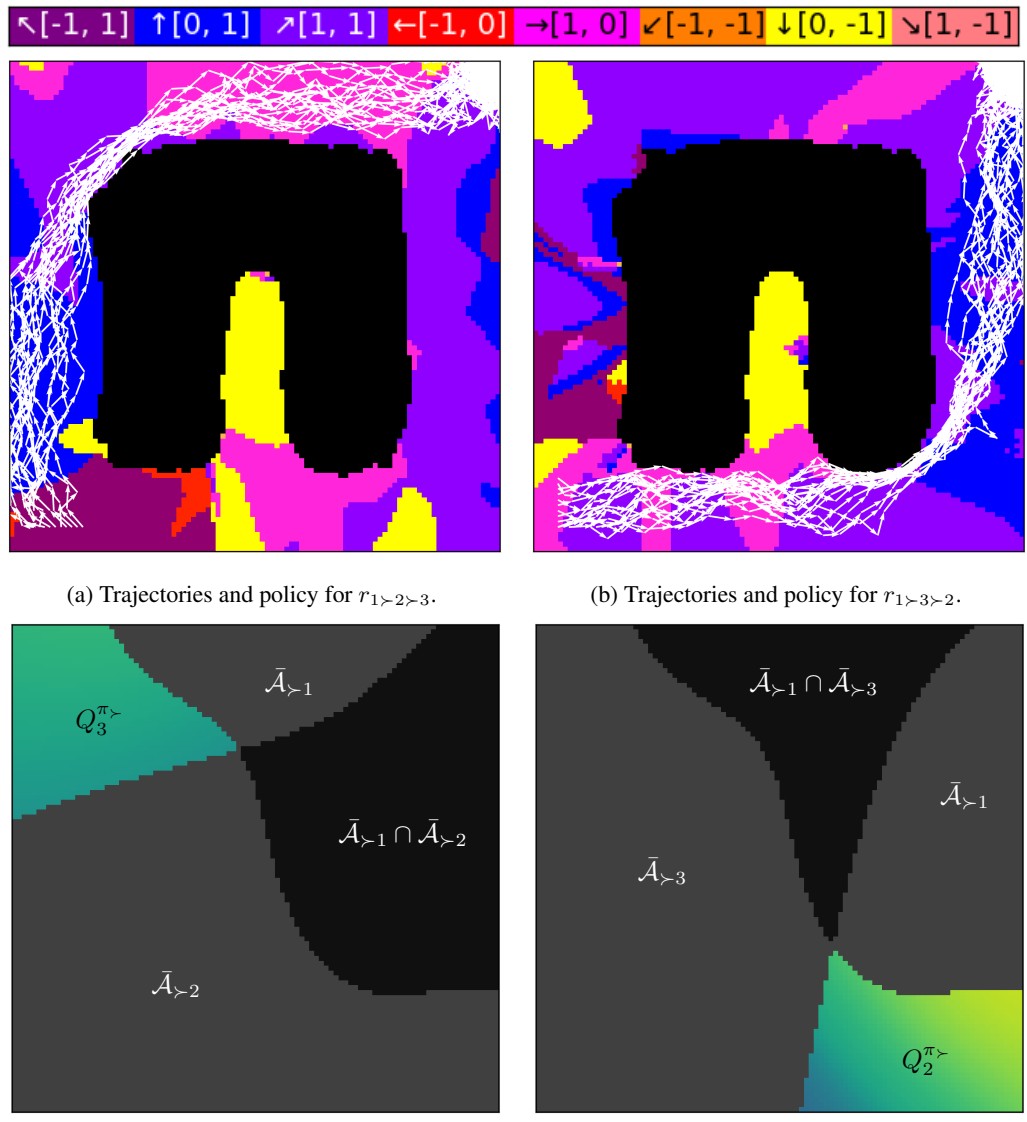

(a) Trajectories and policy for $r_{1 \succ 2 \succ 3}$.

(b) Trajectories and policy for $r_{1 \succ 3 \succ 2}$.

(c) Indifference space and global policy for $r_{1 \succ 2 \succ 3}$.

(d) Indifference space and global policy for $r_{1 \succ 3 \succ 2}$.

Figure 8: Trajectories, policy, action indifference space, and value function for global agents. For the bottom row images, the agent was placed at marker 5 in Fig. 7e (i.e., in the bottom left, close to the starting points of the trajectories). While both tasks are constrained by $\bar{\mathcal{A}}_{\succ 1}$, which prevents the agent from getting too close to the obstacle, the differing prioritization of $r_2$ and $r_3$ induces different global indifference spaces through $\bar{\mathcal{A}}_{\succ 2}$ and $\bar{\mathcal{A}}_{\succ 3}$. The respective lower-priority task expresses its preference in the global indifference space (colored area). As can be seen, the policy in (c) is constrained to actions that move up, due to the optimality constraint on $Q_2$, while in (d), the optimality constraint on $Q_3$ constraints the policy to actions that move right.

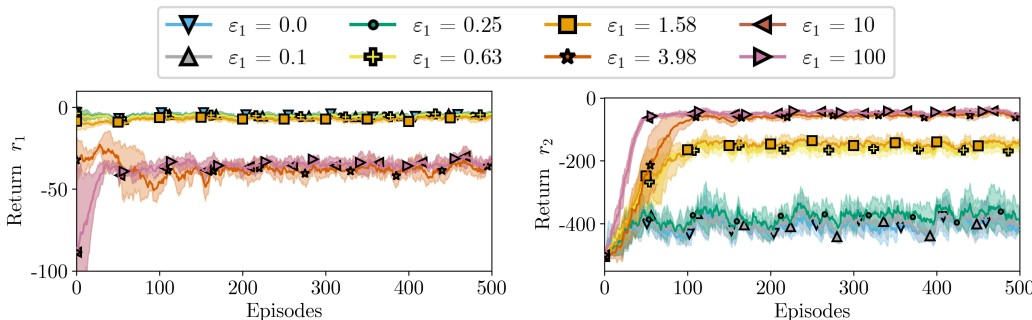

Figure 9: $\varepsilon$ **Ablation** in the 2D navigation environment. **Left**: Episode return of the high-priority obstacle-avoidance task during learning of the low-priority goal-reach task. **Right**: Episode returns of the low-priority task. High-priority subtask performance decreases and low-priority subtask performance increases with larger $\varepsilon$ values.

## H.3   $\varepsilon$ ABLATION

To illustrate how $\varepsilon$ thresholds affect the learned behavior and performance, we perform an ablation study using different values for $\varepsilon_1$. We again use the 2D navigation environment from App. G and assign high priority to the obstacle avoidance tasks and low priority to reaching the top goal. We generate values for $\varepsilon_1$ that are roughly log-spaced between 0 and 100. For this experiment, we pre-train only the high-priority obstacle avoidance task and learn the low-priority goal-navigation agent from scratch. We report the same metrics as for our baseline comparison in Sec. 4.3, meaning during learning of the low-priority tasks $r_2$, we log the episode return of both subtasks, to show the trade-off between the two objectives, depending on the threshold value.

The average results from five different random seeds are shown in Fig. 9, with high-priority task returns shown on the left, and low-priority task returns shown on the right. Performance on the low-priority subtask roughly falls into three groups, depending on $\varepsilon_1$-threshold. With very small values for $\varepsilon_1$, i.e. $\varepsilon_1 = 0.0, 0.1$, or $0.25$, the algorithm can not improve the low-priority subtasks. This is expected because the small thresholds induce such small indifference spaces that none of the available actions allow the agent to improve the performance of the low-priority subtask. At the same time, these small $\varepsilon_1$ thresholds result in optimal performance for the high-priority task, since they allow almost no divergence from the optimal behavior. Independently of the value for $\varepsilon_1$, as can be seen in the left panel of Fig. 9, the episode returns for $r_1$ are roughly constant, since $Q_1^*$ does not change during learning of $r_2$.

On the other end of the range, i.e. with large $\varepsilon_1$ values of $3.98, 10$, or $100$, the algorithm achieves optimal performance for the low-priority task. This is because the large $\varepsilon_1$ values induce indifferences space that are not restrictive enough to prevent the agent from hitting and moving through the obstacle, which is reflected by the high costs that these agents obtain under the high-priority task.

Most interesting are the remaining $\varepsilon_1$ values, namely $\varepsilon_1 = 0.63$ and $\varepsilon_1 = 1.58$. With these values, the agent achieves the same near-optimal performance with respect to the high-priority obstacle avoidance task while also managing to improve low-priority subtask performance considerably. This is because the resulting indifference spaces prevent obstacle collisions (as can be seen in Fig. 7 with $\varepsilon_1 = 1$) but still allow for many actions that can be used to optimize the low-priority task.

In summary, the $\varepsilon_i$ scalars have a strong effect on the behavior and lower-priority subtask performance. Too large thresholds are ineffective at forbidding undesired actions, while too small thresholds are too restrictive and prevent the agent from improving subtask performance at all. However, as described in Sec. 6, it is straightforward to infer these adequate threshold values by analysis of the higher-priority Q-function, even when the action space is of high dimensionality.

# I  REPRODUCIBILITY

A GitHub repository with the implementation of the algorithm, experiment setup with hyperparameters, and documentation is available here: `https://github.com/frietz58/psqd/`. The repository provides the complete PSQD implementation and can be used to reproduce the results in this paper.

