# OpenReview forum: "Prioritized Soft Q-Decomposition for Lexicographic Reinforcement Learning"
_ICLR.cc/2024/Conference — ICLR 2024 poster_

### Official Review · Reviewer_rq81 · 2023-10-26

**Soundness:** 3 good
**Presentation:** 3 good
**Contribution:** 3 good
**Rating:** 6
**Confidence:** 3

**Summary:**

The paper addresses the challenges of lexicographic MORL problems by introducing a novel approach called prioritized soft Q-decomposition. This technique leverages the value function of previously learned subtasks to constrain the action space of subsequent subtasks. The experimental results conducted on both simple and complex scenarios substantiate the efficacy of this method.

**Strengths:**

The paper is well organized and easy to read.
The proposed method, in its simplicity, manages to be effective in tackling the problem.

**Weaknesses:**

The proposed method appears to be sensitive to the parameter ε, and the manual selection of this parameter is non-trivial.
Additionally, the paper falls short in providing a comparative analysis with existing lexicographic MORL algorithms.

**Questions:**

1. The paper employs equation 7 to approximate ε-optimality based on equation 1, but the relationship between the two equations is not very clear. Including more theoretical insights could enhance the paper's rigor.
2. It's worth pondering whether εi should be state-dependent. Is it possible to find a constant εi that precisely represents the task requirements? If not, it might indicate that the action space is overly restricted in some states, hindering exploration in subsequent tasks, or, conversely, that undesired actions cannot be excluded in certain states.

---

> ### Author Response · Authors · 2023-11-17
> **Response to reviewer rq81**
>
> We kindly thank reviewer *SKG7* for their review of our work. In the following, we answer the brought forward criticism and questions.
>
> ### **$\varepsilon$ threshold sensitivity**
>
> *Reviewer rq81 is concerned about the sensitivity of our method with respect to manually selected thresholds.*
>
> Our method is indeed dependent on the manually selected $\varepsilon_i$ thresholds, however, this is the case for all $\varepsilon$-lexicographic MORL methods, since these thresholds are part of the problem definition. In the limitation section of our paper, we suggest multiple approaches to finding adequate values for $\varepsilon_i$ and note these are fundamentally problem-dependent, the same way that a reward function is.
>
> See also our answer "Threshold ablation" to reviewer *tsYf*.
> We have included an additional ablation study with different $\varepsilon_i$ values in the revised manuscript (see Appendix H.3), to further illustrate how these scalars affect performance and agent behavior.
> We further note that we describe how adequate $\varepsilon_i$ thresholds can be selected in Section 6 of the paper.
>
> ### **Comparison to existing lexicographic RL algorithms**
>
> *Reviewer rq81 suggests additional comparison with lexicographic RL algorithms.*
>
> As we write in the related work section, we believe to contribute the first algorithm for **continuous** action-space lexicographic MORL problems. As stated in our answer "8. Discrete PSQD version" to reviewer *SKG7*, a discrete version of our algorithm would be conceptually similar to existing works and we do not consider this comparison informative for learning in continuous spaces.
>
> ### 1. Relationship between Eq.(1) and Eq. (7)
>
> Equation (1) defines a general lexicographic constraint and can also be found in related works on lexicographic RL. Equation (1) simply states that the optimization of subtask $i$ is constrained to a set $\Pi$ of policies, where all policies in $\Pi$ are also (near-) optimal to all higher-priority subtasks ${1, \dots, i-1}$.
>
> Since the explicit computation of $\Pi$ is not practical, especially for probabilistic policies with continuous action spaces, in Eq. (7) we instead make a state-based version of Eq. (1), where the performance measures $J_i$ are the Q-functions. This is also consistent with existing related work on lexicographic RL.
>
> We hope this clarifies the relationship of these two equations.
>
> ### 2. State-dependent $\varepsilon_i$
>
> > It's worth pondering whether εi should be state-dependent.
>
> This is indeed an interesting discussion and an interesting research topic.
>
> However, we want to highlight that, although $\varepsilon_i$ are fixed scalars, the Q-functions to which these thresholds relate are very much state-dependent. In effect, this results in indifference spaces that are very large (i.e. permissive) in some states, while only being restrictive in states *where it really matters*. For example, when far from the obstacle, the lexicographic constraint on obstacle avoidance allows nearly every action, however when close to the obstacle, it forbids the selection of those actions that would lead to obstacle collision (since those actions are clearly sub-optimal for the obstacle-avoidance task).
>
> This can also be seen in Figure 7 in the appendix, where we visualize the agent and the indifference space at multiple locations in the environment. Most of these are relatively close to the obstacle, nevertheless, the indifference space in Subfigure 7.h is much larger and more permissive than in Subfigure 7.d or 7.f, due to varying proximity between the agent and the obstacle.
>
> We hope this addresses reviewer *rq81*'s questions regarding our work and again thank them for their review.

---

> > ### Comment · Reviewer_rq81 · 2023-11-21
> >
> > Thank you for the author's response.
> >
> > I still have some concerns regarding Eq. (7) and Eq. (1). Are they theoretically equivalent? Alternatively, does Eq. (1) yield Eq. (7) with a state-dependent $\epsilon(s)$, , where the function $\epsilon(s)$ is related in some way to the original $\epsilon$ in Eq. (1)?

---

> > > ### Author Response · Authors · 2023-11-21
> > > **Non-equivalence between Eq. (1) and Eq. (7)**
> > >
> > > Eq. (1) and Eq. (7) are not equivalent. We primarily feature Eq. (1) to provide intuition for lexicographic RL as part of the problem definition.
> > >
> > > Strictly speaking, Eq. (1) is a global constraint, where the thresholds $\varepsilon_i$ would be expressed on some global performance criteria like the value of the starting state or the policy's expected return. Eq. (7) is a local and state-based approximation of Eq. (1). Importantly, Eq. (7) is more restrictive, gives us more control over the agent's behavior, and the satisfaction of the constraint in Eq. (7) **implies** satisfaction of the constraint in Eq. (1).  When the lexicographic constraint is enforced in every state, as in Eq. (7), it clearly also holds globally and in expectation.
> > >
> > > We will add the words *global* and *local* to avoid confusion about the equivalent of Eq. (1) and Eq. (7). We note again that the conversation of global Eq. (1) into local Eq. (7) is common practice (see 2022 paper by Skalse, Hammond, Griffin and Abate or 2022 paper by Wray, Tiomkin, Kochenderfer and Abbeel) and especially important for safety-critical applications.

---

### Official Review · Reviewer_SKG7 · 2023-11-04

**Soundness:** 3 good
**Presentation:** 2 fair
**Contribution:** 2 fair
**Rating:** 5
**Confidence:** 4

**Summary:**

This paper proposes an algorithm called prioritized soft Q-decomposition (PSQD) to solve complex lexicographic multi-objective reinforcement learning (MORL) problems. In the setting, n subtasks are prioritized, and the available policy set is reduced as each subtask is optimized in the predefined order. Instead of explicitly representing the available policy sets, the authors consider restricting action space to satisfy the lexicographic order. For implementation, Soft Q-learning (SQL) is adopted to deal with continuous action space. Numerical results show that PSQD performs better than the other baselines in the considered environments, where there are two subtasks.

**Strengths:**

- The authors provide mathematical formulations on the main paper to support the soundness of the algorithm.
- This paper contains a dense appendix for detailed explanations.
- The authors provide extensive study on previous works.

**Weaknesses:**

1. Presentation needs to be improved. While the considered setting - lexicographic MORL - is clear, the flow of the algorithm is hard to understand. High-level pictograms can help the readers understand the content.

2. There is no pseudocode, so I am confused about the implementation of the algorithm. As far as I understand, the proposed algorithm is one of the following:

Candidate 1) For subtask 1 to n-1, run parallel SQL in equation (12). Then restrict action space satisfying epsilon-optimalty of subtasks 1 to n-1. In the restricted action set, run SQL for subtask n in equation (12) and recover optimal policy using (13).

Candidate 2) For subtask 1, run SQL in equation (12). Acquire restricted action space A_1.  Run (12) on A_1 and acquire A_2. ... After acquiring A_{n-1}, run (12) and (13) for subtask n.

Which one is right? Please provide a pseudocode.

3. For clarity of the setting, I want to raise several fundamental discussions regarding the lexicographic setting.
- When lexicographic MORL setting is considered in practice?
- What is the clear difference between constrained (MO)RL?
- Who decides the priority order? Is it valid to assume that we always know the priority order?
- How do orders of subtasks affect the final performance in experiments? If the order is crucial, discussion on setting order is important.
- What if there are tasks that are not prioritized (e.g., "equally" important, or "we do not know")?

4. Experiments deal with only two subtasks. Can the authors show another environment containing more than two subtasks?

5. For reproducibility, it would be better for the authors to provide anonymous source code.

6. There is confusion in eq. (7). Does Q_i in eq. (7) mean the optimally trained one (i.e., Q_i^*)?

**Questions:**

Please check the weakness part. Additional questions are as follows.

7. In number 2 in Weakness, if one of the candidates is PSQD, do the authors think that the other one is also a valid algorithm? (It looks like candidate 1 does not use order information of 1 > ... > n-1).

8. If SQL is used, PSQD can be extended to discrete action space since the integral is changed to summation in equation 5. Then we may compare PSQD with the previous work of Skalse et al (2022).
Also in that discrete action case, do authors think that action set refinement is still valid?

9. Confusing notation in eq (1). J_0, epsilon_0, Pi_0 is not explicitly defined.

---

> ### Author Response · Authors · 2023-11-17
> **Response to reviewer SKG7**
>
> We kindly thank reviewer *SKG7* for their review of our work. In the following, we answer the brought forward criticism and questions.
>
> ### 1. & 2.
>
> We fully agree with reviewer SKG7 that pseudocode and pictograms will benefit understandability.
>
> Therefore, in the revised version of the manuscript in Appendix "D PSQD algorithm details", we have added algorithm pseudocode for the pre-training step, the adaptation step, a pictographic overview of our framework (Fig. 5), and additional details regarding our method. We will release our full codebase once the paper is accepted, which will benefit the understandability and reproducibility of our work.
>
> ### 3
> + 3.1 We believe lexicographic MORL has numerous benefits over traditional MORL which relies on scalarization: Lexicographic MORL algorithms allow for the transfer of simple subtask agents to complex lexicographic problems, benefits interpretability by decomposition, and offer constraint-satisfaction guarantees that can make for a safe exploration framework. Our method is for continuous action-space MDPs, unlike existing lexicographic RL methods for discrete action-space MDPs. We make an important, fundamental contribution by extending lexicographic MORL to continuous action-space MDPs, which have many applications in real-world problems, like robot control.
>
> +  3.2 Lexicographic MORL can be seen as a special form of constrained MORL, where the constraints are of a lexicographic nature and for a chain-like directed, acyclic graph. This is not the same as, e.g. constrained policy optimization (Achiam et al, 2017), which places multiple constraints on the policy that are not lexicographically ordered. Furthermore, in lexicographic MORL, the constraints are based on the learned subtask solutions (e.g. Q-functions) and not based on manually defined functions, as is usually the case in constrained MORL.
>
> +  3.3 The RL practitioner/designer decides on priority order. Priority order is part of the problem definition, in the same way that defining the MDP (state-space, action-space, reward function) is. Defining a lexicographic RL problem requires a priority order, in the same way that defining a regular RL problem requires a reward function. If the RL practitioner can not decide the order of two subtasks, it is possible to treat those subtasks with equal priority by simply summing those corresponding reward functions. We discuss this more in our response below to question 3.5.
>
> + 3.4 We have experiments with three subtasks in Appendix G.2. These experiments show that changing the priority order changes the resulting behavior, which is the expected result. Changing the priority order changes the problem definition, therefore comparing the performance of multiple such agents does not make sense. Related to this is also our new ablation study on the priority thresholds in Section H.3 in the appendix.
>
> + 3.5 If tasks are of equal importance, then it is valid to sum their reward functions. We can then learn a single Q-function $Q_{1+2}$ for $r_1 + r_2$. We can use $Q_{1+2}$ like any subtask Q-function and define a corresponding $\varepsilon_{1+2}$ threshold for this Q-function. Keep in mind, however, that simply adding the subtask reward functions can produce unexpected behavior, especially when these subtasks Q-functions are semantically incompatible. We argue that practitioners should consistently order tasks by priority. If the tasks are compatible, then the lexicographic constraints do not hurt performance, and if they are incompatible, we have performance guarantees for the higher-priority tasks, which is not the case for MORL methods that don't use priority.
>
> ### 4.
> We have experiments that use more than two subtasks in Appendix G.2.
>
> ### 6.
>
> $Q_i$ in Eq. (7) refers to the subtask Q-functions of an agent that is lexicographically optimal for all $n-1$ higher-priority tasks. These are not optimal Q-functions $Q_1^*, \dots, Q_n^*$. We have changed the sentence preceding Eq. (7) to emphasize this.
>
> ### 8.
>
> As discussed in the introduction and related work section, the primary contribution of our paper is an algorithm for *continuous* action-space MDPs. A discrete version of PSQD would be conceptually similar to existing methods like value-based lexicographic MORL (Skalse et al. 2022), lexicographic value iteration (Wray et al. 2015), or TLO (Zhang et al, 2023), which all require finite action spaces.
>
> ### 9.
>
> This is the standard lexicographic constraint consistent with related work. As we write in the paper $J_i$ are some performance measures (like value functions, Q-functions), the symbol $\pi$ is defined as policy, and the symbol $\varepsilon_i$ is defined, in the next line, a performance threshold scalar for subtask $i$.
>
> We again thank reviewer *SKG7* for their review of our work, which we believe allowed us to improve the understandability and reproducibility of our work, due to the added pseudocode and picographic overview.

---

### Official Review · Reviewer_tsYf · 2023-11-06

**Soundness:** 3 good
**Presentation:** 2 fair
**Contribution:** 3 good
**Rating:** 6
**Confidence:** 4

**Summary:**

This paper presents a new methods to learn Reinforcement Learning policies that obey lexicographic subtask constraints. To make this efficient, the presented method creates zero shot Q-functions and strategies for the priority constrained task by composing transformed versions of the individual subtasks through their limitation to the indifference space of actions for the higher priority tasks and the transformation of the reward and Q value function to infinitely penalize such action choices. The resulting one shot version of each task value function and policy can then be adapted offline or online to achieve potentially near-optimal performing policies.
The main contributions in the paper are in the novel methodology and corresponding learning algorithm to form RL strategies by composing subtask functions such as to obey priority constraints expressed in the value function of the higher level tasks.

**Strengths:**

The paper tries to address two very important problems in Reinforcement Learning for control tasks in: i) providing an effective means of composing overall behavior from learned subtasks without significant need for new data collection and re-learning, and ii) to enforce strict priority constraints during composition as well as subsequent learning to optimize from the initial one-shot policy. These abilities are very important in the area of robot control in order to be able to enforce learned safety and performance constraints when new task compositions have to be learned.
The approach seems overall sound (although the description is lacking in a few places) and the results demonstrate that the method can form both single-shot strategies and more optimized policies using additional off-line (or on-line training) training that maintains the highest priority constraint.

**Weaknesses:**

The main weaknesses of the paper are in a lack of discussion of the full range of situations in the presentation of the underpinnings of the framework and in the incomplete description of the experiments presented in the paper.
The former here seems the biggest weakness and mainly relates to the complete absence of a discussion and consideration of cases where subtasks might not be compatible and thus the indifference space of actions might become empty. It would be very important for the authors to discuss this situation and how the algorithm would react under those circumstances. This is even more important as in the description of the decomposed learning task only the highest and the lowest priority subtasks are discussed  (where in the case of contradictory tasks, the lowest priority task would no longer have a usable actions space and could therefore not have a policy).

In terms of the experiment presentation, it would be very useful if the main paper could contain at least a basic description of the environment, the action space, and in particular the subtasks and corresponding reward functions. For the latter (the reward functions), even the Appendices do not seem to contain more than an rough description of the principles of the reward function of the obstacle subtask. It would be very important for reproducability but also for a better understanding of the reader if the authors were to include the exact reward function for each subtask as well as the \epsilon thresholds that were used for the experiments presented in the paper (and these should be in the main paper).

Another slight weakness is that while the paper indicates that the pick of thresholds \epsilon is difficult, it does not provide any analysis of this. A brief ablation in term of \epsilon for the obstacle task in the 2D navigation experiment would have been very useful, as would be a brief discussion how such thresholds might be picked and what the tradeoffs of different picks are.

**Questions:**

The main questions arise out of the weaknesses stated above:
How does the proposed approach deal with incompatible subtasks ? Does it simply eliminate all tasks with empty indifference action sets for higher priority tasks and then operate in the same way as presented in the paper ?

How sensitive is the approach to the specific choice of \epsilon thresholds ? Is there a way that an ablation could be performed that would investigate the sensitivity of the top priority task's threshold in the navigation experiments ?

---

> ### Author Response · Authors · 2023-11-17
> **Response to reviewer tsYf**
>
> We kindly thank reviewer *tsYf* for their review of our work. In the following, we answer the brought forward criticism and questions.
>
> ### **Reproducibility and reward functions**
>
> *Reviewer tsYf is concerned that environments and reward functions are not sufficiently described.*
>
> Initially, we did not explicitly state the reward functions because, for reproducibility reasons, we plan to publish our complete codebase alongside the camera-ready version of the paper, which contains the reward functions and the complete environment specification. For completeness and to address this concern, we have now detail the reward functions in section G.1 in the appendix.
>
> We hope this eliminates the concern w.r.t reproducibility.
>
> ### **Discussion on incompatible subtasks**
>
> *Reviewer tsYf wishes for a more thorough discussion on situations where subtasks are incompatible and is concerned that our proposed method might fail in such cases since they believe that the indifference spaces would become empty.*
>
> We fully agree that this discussion is important, interesting, and benefits the understandability of our method. To address this point, we have added a new discussion section, based on the following points, to the revised manuscript in Appendix E.
>
> - Firstly, when subtasks are *semantically* incompatible (e.g. when $r_2 = -r_1$, where $r_1$ rewards going to the left and $r_2$ rewards going to the right), it should be recognized that the MORL task is fundamentally ill-posed since no agent can behave optimally, at the same time, for two objectives that are entirely incompatible. This inherent challenge exists independently of the optimization method.
> - Secondly, we want to highlight an advantage of our method over MORL methods without task priorities: If two objectives are semantically incompatible, our agent attempts to solve the lower-priority task as best as possible, while the constraint ensures that the higher-priority task is still solved optimally (up to the $\varepsilon$ threshold). If the lower-priority task rewards driving to the left, but the higher-priority task rewards driving to the right, our agent *has* to drive to the right, due to the lexicographic constraint. A non-lexicographic MORL agent will instead behave unpredictably in an attempt to maximize the incompatible subtasks at the same time (see Russel & Zimdars 2003 paper "Q-Decomposition for RL agents").  In fact, semantically incompatible subtasks were precisely the original motivation for lexicographic MORL (Multi-criteria Reinforcement Learning, 1998, Gábor, Kalmár and Szepesvári).
> - Lastly, with our method, the indifference-space can never be empty (although very small). This is due to our problem definition (threshold *relative* to subtask Q-function) and the iterative nature of our learning algorithm. By first adapting the second-highest subtask solution using the lexicographic constraint of the highest-priority subtask, the second-highest priority subtask solution only assigns high value to actions that are in the highest priority subtask's indifference space. Thus, after adaptation, the indifference spaces of subtasks overlap, even when the original subtask rewards are semantically incompatible. Generally speaking, for some priority level $i$, in our adaptation step, all **higher-priority** subtask solutions are already "compatible" because they have already been adapted. Then, even in the extreme case where we set $\varepsilon_i = 0$, all **lower-priority** tasks can still "chose" from all optimal actions for task $r_i$. If, on the other, *absolute* performance thresholds were used that are not relative to the (adapted) subtask Q-functions, the intersection of those indifference spaces could indeed be empty.
>
> ### **Threshold ablation**
>
> *Reviewer tsYf wonders about the sensitivity of our method to differing $\varepsilon_i$ thresholds and suggests an ablation study.*
>
> The behavior that our agent learns is indeed governed by the $\varepsilon_i$ thresholds since these thresholds give rise to the action indifference space for each subtask. Intuitively, **larger** $\varepsilon_i$ values mean that the performance for task $r_i$ is allowed to **degrade more**, in favor of increased performance for lower-priority tasks. Conversely, lower $\varepsilon_i$ values mean that the performance for task $r_i$ must remain **more optimal**, while lower-priority tasks are more restricted in their subtask optimization.
> We discuss in Section 6 of the paper how adequate values for $\varepsilon_i$ can be found practically.
>
> To address this concern, we have conducted and added an ablation study on different $\varepsilon$ values in the appendix, Section H.3.
> This experiment empirically confirms the above-described relationship between $\varepsilon_i$ scalars and subtask performance.
>
>
>
> We again thank reviewer *tsYf* for their review of our work, we firmly believe the review allowed us to add important and beneficial content to our paper.

---

### Author Response · Authors · 2023-11-17
**Global response**

We thank all reviewers for their constructive and valuable feedback.

Overall, the reviewers find our contribution very important, effective in tackling lexicographic MORL problems, and mathematically sound.

The main points of criticism are on presentation and clarity, reproducibility, and missing discussion of certain cases.
In the revised version of the manuscript we address these points by

+ describing our learning algorithm in greater detail (appendix section D),
+ adding pseudocode and a pictographic overview of our method (appendix section D),
+ adding a discussion section on semantically incompatible subtasks (appendix section E),
+ and adding an ablation study on threshold scalars $\varepsilon_i$ (appendix section H.3).

With respect to reproducibility, we note that we will release the full codebase on GitHub, alongside the camera-ready version of the paper.

Please see our individual responses for more details.

---

### Comment · Area_Chair_VtxZ · 2023-11-20
**Author-Reviewer Discussion Period Ending November 22**

Hi,

Thanks for your help with the review process!

There are only two days remaining for the author-reviewer discussion (November 22nd). Please read through the authors' response to your review and comment on the extent to which it addresses your questions/concerns.

Best,\
AC

---

### Meta-Review · Area_Chair_VtxZ · 2023-12-12

**Metareview:**

The paper considers the ability to improve the effectiveness with which reinforcement learning (RL) can be applied to complex tasks by formulating these tasks in terms primitive subtasks and reusing their solutions. The paper proposes prioritized soft Q-decomposition (PSQD), an policy learning algorithm for lexicographic multi-objective reinforcement learning (MORL) problems that consist of prioritized subtasks. Given a set of prioritized subtasks, the method reduces the available policy set as each subtask is optimized in a predefined order. Rather than explicitly representing the available policy sets, the authors restrict the action space to satisfy the lexicographic order. The framework is implemented using soft Q-learning in order to deal with the continuous action space. Experiments involving both simple and complex domains demonstrate that PQSD outperforms contemporary baselines.

The paper addresses two important RL problems, notably the ability to effectively compose learned subtasks without the need for an extensive amount of additional data or additional learning, and the ability to enforce strict priority constraints during skill composition. As several reviewers point out, the proposed approach (PSQD) is sound and the results reveal that it is capable of formulating both single-shot policies as well as more optimized policies using additional training that maintains the highest priority constraint. The primary concerns that the reviewers initially raised related to the lack of discussion regarding cases for which subtasks may not be compatible; insufficient experimental details; issues with clarity due in part to the lack of pseudocode; and the potential need to compare to existing lexicographic MORL algorithms. The authors address most of these concerns in their responses to the reviewers (though the AC does not see clarification regarding the details of the experimental setup other than the rewards).

Note: While Reviewer SKG7 did not reply to the authors' response to their initial review, they did increase their score from a 3 (Reject) to a 5 (Marginally Below the Acceptance Threshold). Reviewer tsYf did not respond despite several efforts on the part of the authors and the AC.

**Justification For Why Not Higher Score:**

While the authors' response addressed several of the reviewers' initial questions/concerns, as suggested by at least one reviewer increasing their score (Reviewer SKG7 from a 3 to a 5), the opinion of the AC is that the contributions do not warrant a higher score.

**Justification For Why Not Lower Score:**

The AC is erring on the positive side, but would not be opposed to a lower score given the lack of enthusiasm among the reviewers.

---

### Decision · Program_Chairs · 2024-01-16

Accept (poster)